# Molecular Simulations with in-deMon2k QM/MM, a Tutorial-Review [note 1]

**DOI:** 10.3390/molecules24091653

**Published:** 2019-04-26

**Authors:** Aurélien de la Lande, Aurelio Alvarez-Ibarra, Karim Hasnaoui, Fabien Cailliez, Xiaojing Wu, Tzonka Mineva, Jérôme Cuny, Patrizia Calaminici, Luis López-Sosa, Gerald Geudtner, Isabelle Navizet, Cristina Garcia Iriepa, Dennis R. Salahub, Andreas M. Köster

**Affiliations:** 1Laboratoire de Chimie Physique, CNRS, Université Paris Sud, Université Paris Saclay, 15 avenue Jean Perrin, 91405 Orsay, France; aurelio.alvarez-ibarra@u-psud.fr (A.A.-I.); karim.hasnaoui@idris.fr (K.H.); fabien.cailliez@u-psud.fr (F.C.); xiaojing.wu@u-psud.fr (X.W.); 2CNRS Laboratoire de Biochimie Théorique, Institut de Biologie Physico-Chimique, PSL University, 75005 Paris, France; 3Matériaux Avancés pour la Catalyse et la Santé, UMR 5253 CNRS/UM/ENSCM, Institut Charles Gerhardt de Montpellier (ICGM) Montpellier CEDEX 5, 34090 Montpellier, France; Tzonka.Mineva@enscm.fr; 4Laboratoire de Chimie et Physique Quantiques, IRSAMC, Université Paul Sabatier, 118 Route de Narbonne, 31062 Toulouse CEDEX 4, France; erome.cuny@irsamc.ups-tlse.fr; 5Programa de Doctorado en Nanociencias y Nanotecnología, CINVESTAV, Av. Instituto Politécnico Nacional, 2508, A.P. 14-740, Ciudad de México 07000, Mexico; pcalamin@cinvestav.mx; 6Departamento de Química, CINVESTAV, Av. Instituto Politécnico Nacional, 2508, A.P. 14-740, Ciudad de México 07000, México; llopezs@cinvestav.mx (L.L.-S.); geudtner@cinvestav.mx (G.G.); 7Laboratoire Modélisation et Simulation Multi Échelle, Université Paris-Est, MSME, UMR 8208 CNRS, UPEM, 5 bd Descartes, 77454 Marne-la-Vallée, France; isabelle.navizet@u-pem.fr (I.N.); cristina.garciairiepa@u-pem.fr (C.G.I.); 8Department of Chemistry, Centre for Molecular Simulation, Institute for Quantum Science and Technology and Quantum Alberta, University of Calgary, 2500 University Drive N.W., Calgary, AB T2N 1N4, Canada; dennis.salahub@ucalgary.ca; 9College of Chemistry and Chemical Engineering, Henan University of Technology, No. 100, Lian Hua Street, High-Tech Development Zone, Zhengzhou 450001, China

**Keywords:** QM/MM simulations, DFT, electron and nuclear dynamics

## Abstract

deMon2k is a readily available program specialized in Density Functional Theory (DFT) simulations within the framework of Auxiliary DFT. This article is intended as a tutorial-review of the capabilities of the program for molecular simulations involving ground and excited electronic states. The program implements an additive QM/MM (quantum mechanics/molecular mechanics) module relying either on non-polarizable or polarizable force fields. QM/MM methodologies available in deMon2k include ground-state geometry optimizations, ground-state Born–Oppenheimer molecular dynamics simulations, Ehrenfest non-adiabatic molecular dynamics simulations, and attosecond electron dynamics. In addition several electric and magnetic properties can be computed with QM/MM. We review the framework implemented in the program, including the most recently implemented options (link atoms, implicit continuum for remote environments, metadynamics, etc.), together with six applicative examples. The applications involve (i) a reactivity study of a cyclic organic molecule in water; (ii) the establishment of free-energy profiles for nucleophilic-substitution reactions by the umbrella sampling method; (iii) the construction of two-dimensional free energy maps by metadynamics simulations; (iv) the simulation of UV-visible absorption spectra of a solvated chromophore molecule; (v) the simulation of a free energy profile for an electron transfer reaction within Marcus theory; and (vi) the simulation of fragmentation of a peptide after collision with a high-energy proton.

## 1. Introduction

The hybrid quantum mechanical/molecular mechanical (QM/MM) scheme is a computationally efficient approach to simulate physicochemical phenomena with adaptive levels of accuracy [1,2,3]. As pointed out by Warshel, Karplus and Levitt in the 1970s [1,2,3], processes involving key modifications of electronic structures must be modeled by quantum mechanical approaches while the environment can be well approximated by classical molecular mechanical approaches. For example, the breaking or the formation of covalent bonds involves a rather restricted number of atoms even though they are modified by the electrostatic atmosphere created by the environment. The promise of QM/MM methodologies is to retain a quantum-mechanical description only for atoms that require QM, while describing the environment with a molecular-mechanics force field. Since the pioneering studies from the 1970s [1,2,3], continuous progress has been made. We refer interested readers to excellent review papers [4,5] and to other articles of this special issue of *Molecules*.

Computer software implementing QM/MM are widespread. Some of them such as Cuby [6], PUPIL [7], Chemshell [8], QMMM [9] or LICHEM [10] involve interface between QM and MM packages. Other computer codes implement both QM/MM in unique programs without the need for interfaces. This is, non-exhaustively, for instance, the case of AMBER [11], NWCHEM [12], QCHEM [13], Gaussian [14] or deMon2k [15].

deMon2k is a program specialized in Kohn–Sham [16] density functional theory (DFT) within the Auxiliary DFT framework [17]. The latter permits remarkably fast evaluation of energies, potentials and properties. It is parallelized within the *Message Passing Interface* framework [18]. Therefore, it is a very promising basis for conducting hybrid QM/MM simulations with DFT as the electronic-structure method. This article follows another one written in 2015 [19] by some of us and other colleagues on the general possibilities offered by deMon2k to perform QM/MM simulations [15]. In this article we focus on the internal QM/MM module that does not involve an interface with an external MM package but evaluates all energy terms internally. Since 2015 several new capabilities have been incorporated into the program. These are listed in Table 1. The program now permits QM/MM simulations with either non-polarizable or polarizable force fields and includes a polarizable continuum model for remote environments. Link atoms [20] and capping potentials have been incorporated to deal with QM/MM frontiers that cut covalent bonds. Both Born–Oppenheimer molecular dynamics (MD) simulations [21,22,23] and Ehrenfest non-adiabatic MD simulations are now available. deMon2k also provides one of the very few implementations for conducting attosecond electron dynamics within polarizable MM environments [24]. A first aim of this article is to provide users with a consistent description of the QM/MM methodology implemented in deMon2k at the dawn of 2019. This is the object of Part I.

QM/MM simulations are complex to set up because of the intrinsic technicity of this kind of simulations. A second objective of this article is to provide user-friendly examples to learn how to use deMon2k in practice in QM/MM simulations. In part II, six projects illustrating some of the QM/MM capabilities of the program are presented. For each of them, a tutorial encompassing delineated computational protocols, guidelines to prepare program inputs and explanation on how to extract useful information from output files are deposited on the deMon2k website [28]. The deMon2k version used for this tutorial paper is a developer version which is based on the recently released public version of deMon2k (5.0). All versions are freely accessible to academic groups and we refer to the deMon2k website for details on how to obtain a license.

## 2. The in-deMon2k Implementation of Quantum Mechanical/Molecular Mechanical (QM/MM)

### 2.1. General Framework for Additive QM/MM

We detail in this section the QM/MM formalism implemented in deMon2k. Throughout this article i and j will refer to electrons or corresponding orbitals, A and B to nuclei from the QM region and K and L to atoms from the MM region. For simplicity, the Greek symbols μ and ν will be used to index atomic orbitals and simultaneously label the corresponding basis functions. A similar indexing and labeling will be used for the atom-centered auxiliary functions. Here $\bar{k}$ and $\bar{l}$ denote primitive Hermite Gaussian auxiliary functions. Vectors are written in bold, $R_{XY}$ refering to the vector between X and Y. Vectors with only one or no sub index refer to position vectors.

#### 2.1.1. Hybrid QM/MM with Non-Polarizable Force Fields

We follow an additive formalism to express the potential energy. The total QM/MM energy ($E^{QM/MM}$) is given by Equation (1). It is the sum of three terms, namely the energy of the QM region ($E^{QM}$), the energy of the MM region ($E^{MM}$) and the interaction energy between the QM and MM regions ($E^{QM*MM}$). $E^{QM}$ is calculated within the framework of Kohn–Sham density functional theory (DFT) [16].
(1)$$E^{QM/MM}\left[\rho \right]=E^{QM}\left[\rho \right]+E^{MM}+E^{QM*MM}\left[\rho \right]$$
(2)$$E^{QM}\left[\rho \right]=T_{s}\left[\rho \right]+\;\int v_{ext}\rho \left(r\right)dr+J\left[\rho \right]+E_{xc}\left[\rho \right]+E_{NN}$$

In Equation (2) $T_{s}$ is the kinetic energy of the non-interacting reference system usually expressed in terms of the Kohn–Sham orbitals, $v_{ext}$ is the external potential created by the nuclei of the QM region, J is the classical Coulomb repulsion between the electrons and $E_{xc}$ is the exchange correlation energy. The square brackets indicate functional dependencies from the Kohn–Sham density, ρ(r), which is calculated from the Kohn–Sham orbitals. The last term in Equation (2), $E_{NN}$, is the classical electrostatic repulsion between the QM nuclei. The actual calculations of the quantum mechanical parts of $E^{QM/MM}\left[\rho \right]$ in deMon2k will be discussed in Section 2.2. The energy of the MM region is obtained with standard non-polarizable force fields (AMBERff99 [29], CHARMM22 [30,31,32] or OPLS-AA [33] are currently implemented). The MM energy is split into two contributions, namely $E^{bonded}$ and $E^{non-bonded}$ collecting respectively the energy terms among bonded MM atoms and among non-bonded MM atoms.

(3)$$E^{MM}=E^{bonded}+E^{non-bonded}$$

(4)$$E^{bonded}=E^{bond}+E^{angle}+E^{torsion}+E^{improper}$$

(5)$$E^{non-bonded}=E^{elec}+E^{LJ}$$

$E^{bonded}$ includes bond energies ($E^{bond}$), bond-angle energies ($E^{angle}$), torsion-angle energies ($E^{torsion}$) and improper dihedral-angle energies ($E^{improper}$). $E^{non-bonded}$ includes the electrostatic energy among charged MM atoms, which is calculated with Coulomb’s law, and a Lennard–Jones energy ($E^{LJ}$) to account for van der Waals interactions. Finally, the interaction between QM and MM regions is obtained following an electrostatic embedding scheme. The interaction energy ($E^{QM*MM}$) reads:(6)$$E^{QM*MM}\left[\rho \right]=E^{\rho q}+E^{Zq}+E^{LJ}$$

(7)$$E^{QM*MM}\left[\rho \right]=-\overset{MM}{\underset{K}{\sum }}\int \frac{q_{K}.\rho \left(r\right)}{\left|r-R_{K}\right|}dr+\overset{QM}{\underset{A}{\sum }}\overset{MM}{\underset{K}{\sum }}\left[\frac{q_{K}.Z_{A}}{\left|R_{AK}\right|}+\varepsilon_{AK}\left[{\left(\frac{r_{AK}^{min}}{R_{AK}}\right)}^{12}-2{\left(\frac{r_{AK}^{min}}{\left|R_{AK}\right|}\right)}^{6}\right]\right]$$

$E^{\rho q}$ is the electrostatic interaction energy between the electron density of the QM region (ρ) and the atomic charges of the MM region ($q_{K}$). $E^{Zq}$ is the Coulomb interaction energy between MM atomic charges and nuclei of the QM region that hold charges $Z_{A}$. $E^{LJ}$ is the sum of Lennard–Jones interactions between MM and QM atoms. $r_{AK}^{min}$ is the distance at which the potential reaches a minimum value and $\varepsilon_{AK}$ is the depth of the potential well. Both are force-field parameters. Since $E^{QM*MM}$ depends explicitly on ρ the Kohn–Sham external potential incorporates the electrostatic influence of the MM region.

#### 2.1.2. Hybrid QM/MMpol with Polarizable Force Fields

When the physical-chemical processes taking place in the QM region involve charge displacement, for example during electron or proton transfers, it becomes essential to capture the mutual polarization of the QM and MM regions [1,34]. The rotational and translational polarization of the molecules pertaining to the MM region atoms are accounted for by molecular dynamics simulations. The remaining electrostatic polarization (induction) needs also to be incorporated in the potential energy. This can be achieved by so-called polarizable QM/MMpol schemes [1]. deMon2k implements a charge-dipole model of induction [24]. This model underlies, for instance, the POL3 model [35] for water and the Amberff02 [36] force field. Each MM site *K* holds an atomic isotropic polarizability ($\alpha_{K}$) which enables the calculation of an induced dipole based on the knowledge of the electric field on the atom ($F_{K}$).

(8)$$\mu_{K}=\alpha_{K}F_{K}=\alpha_{K}\left(F_{K}^{\left(0\right)}+F_{K}^{ind}+F_{K}^{QM}\right)$$

$\alpha_{K}$ is a force field parameter. $F_{K}$ includes three terms, namely, the field created by other permanent MM charges ($F_{K}^{\left(0\right)}$), the field created by other MM induced dipoles $(F_{K}^{ind}$) and that created by the QM region. The latter is itself decomposed into the field arising from the electron cloud ($F_{i}^{\rho }$) and from the nuclei of the QM region ($F_{K}^{ZQM}$). The mathematical expression for each term is given by Equations (9)–(12) [34].

(9)$$F_{K}^{\left(0\right)}=\overset{MM}{\underset{\begin{array}{c} L\\ L\neq K \end{array}}{\sum }}\frac{q_{L}}{r_{KL}{}^{3}}r_{KL}$$

(10)$$F_{K}^{ind}=-\overset{MM}{\underset{\begin{array}{c} L\\ L\neq K \end{array}}{\sum }}T_{KL}\mu_{L}$$

(11)$$T_{KL}=\frac{f_{e}}{r_{KL}^{3}}I-\frac{3f_{t}}{r_{KL}^{5}}\left[\begin{array}{ccc} x^{2} & xy & xz\\ yx & y^{2} & yz\\ zx & zy & z^{2} \end{array}\right]$$

(12)$$F_{K}^{QM}=F_{K}^{Z}+F_{K}^{\rho }=\overset{QM}{\underset{A}{\sum }}\frac{Z_{A}}{r_{KA}{}^{3}}r_{KA}-\int \frac{\rho \left(r\right)}{{\left|r_{K}-r\right|}^{3}}\left(r_{K}-r\right)dr$$

$T_{KL}$ is the polarizability tensor and I is the identity matrix. $f_{e}$ and $f_{t}$ are two damping functions that are introduced to avoid a polarization catastrophe which is a divergence of the polarization energy [37,38]. deMon2k implements three damping models (i.e., linear, exponential and Tinker-like models). The program also offers the possibility to introduce cutoffs beyond which the contribution from induced dipoles $\mu_{L}$ are neglected in the calculation of $F_{K}^{ind}$. The QM/MMpol energy is decomposed into three terms:(13)$$E^{QM/MMpol}\left[\rho \right]=E^{QM}\left[\rho \right]+E^{MMpol}+E^{QM*MMpol}\left[\rho \right]$$

$E^{QM}$ is similar to the QM/MM scheme with a non-polarizable force field. On the other hand, the energy of the MM region now includes a supplementary term reflecting the interaction between induced dipoles and MM charges ($E^{indMM}$).

(14)$$E^{non-bonded}=E^{elec}+E^{LJ}+E^{indMM}$$

(15)$$E^{indMM}=-\frac{1}{2}\overset{MM}{\underset{K}{\sum }}\mu_{K}.F_{K}^{\left(0\right)}$$

The QM/MM interaction energy includes supplementary terms compared to non-polarizable QM/MM:(16)$$E^{QM*MM}\left[\rho \right]=E^{\rho q}+E^{Zq}+E^{LJ}+E^{\rho \mu }+E^{Z\mu }$$

(17)$$E^{\rho \mu }=\frac{1}{2}\overset{MM}{\underset{K}{\sum }}\int \frac{\rho \left(r\right)}{{\left|r-\;R_{K}\right|}^{3}}\mu_{K}.\left(r-\;R_{K}\right)\;dr$$

(18)$$E^{Z\mu }=-\frac{1}{2}\overset{MM}{\underset{K}{\sum }}\mu_{K}.F_{K}^{Z}$$

$E^{\rho \mu }$ and $E^{Z\mu }$ are the induction energies arising from the interaction of induced dipoles with the electron density and the nuclei of the QM region, respectively. The QM/MMpol scheme accounts for the mutual polarization between the QM and MM regions. On one hand, the QM region contributes to the amplitude of the induced dipoles through the electric field $F_{K}^{QM}$. On the other hand, the induced dipoles are incorporated into the embedding Kohn–Sham Hamiltonian. This makes the self-consistent field (SCF) slightly more complicated because, contrary to the QM/MM scheme, the embedding Hamiltonian implicitly depends on the electron density through the induced dipoles ($\mu_{K}$) and has to be recomputed at every SCF cycle. However, this is usually not a major issue in terms of computational performance and SCF convergence.

#### 2.1.3. Long-Range Interactions

deMon2k currently does not implement periodic boundary conditions. Long range effects can be modelled by an Onsager Self-Consistent Reaction Field (SCRF) method to account for a polarizable continuum solvent medium. A recommended QM/MM set-up for large molecular systems is represented in Figure 1. The QM region treated by DFT is delimited by the red circle, the MM region is shown in grey. The reaction field energy is given by [39]:(19)$$E_{RF}=-\frac{1}{2}g\mu^{2}$$
where g is the response function of the dielectric solvent medium with respect to the potential of the QM/MM system and μ is the dipole of the QM/MM system. For spherical cavities the response function reads:(20)$$g=\frac{2\left(\varepsilon -1\right)}{\left(2\varepsilon +1\right)a_{0}^{3}}$$
where ε  is the dielectric constant of the medium and $a_{0}$ the radius of the cavity. $E_{RF}$ is added to the QM/MM energy. If the simulated system is electrically charged a Born term is added to the potential energy: $E_{Born}=\frac{1}{2}\left(1-1/\varepsilon \right)\left(q_{sol}^{2}/a_{0}\right)$, where $q_{sol}$ is the charge of the solute. To avoid entropy-driven diffusion of external atoms into the outer continuum, soft harmonic restraints can be applied on the position of atoms belonging to an outer-layer (Figure 1).

#### 2.1.4. Frontier Interactions

To deal with QM/MM frontiers that cut covalent bonds, link atoms [20] can be employed to saturate the valence of frontier QM atoms. Link atoms are built automatically by deMon2k only if users specify that covalent bonds exist between a pair of QM and MM atoms. Our implementation of link atoms follows most common prescriptions [20,40]. The position of a link atom (L) is uniquely defined by the distance ratio R: $R\equiv d\left(X_{QM}-L\right)/d\left(X_{QM}-Y_{MM}\right)$) where $X_{QM}$ and $Y_{MM}$ are the frontier QM and MM atoms. This way of adding a link atom does not introduce supplementary nuclear degrees of freedom [41]. Link atoms hold basis functions, and eventually tuned pseudo-potentials. Supplementary MM terms are added to treat interaction across the frontier as prescribed in Reference [40] (see Table II of this referenced article).

### 2.2. Density Functional Theory with deMon2k

#### 2.2.1. DFT with Variational Density Fitting

The Kohn–Sham formulation of DFT [16,42] introduces orbitals for a non-interacting reference system in order to avoid the explicit calculation of the kinetic energy functional in Equation (2). These so-called Kohn–Sham orbitals are obtained from the following minimization procedure [43,44]:(21)$$T_{s}\left[\rho \right]=\underset{\Psi \to \rho }{\min}\langle \Psi |\hat{T}\Psi \rangle$$

The wave function, Ψ, of the non-interacting system is expressed by a Slater determinant composed of orthonormal Kohn–Sham orbitals $\psi_{i}\left(r\right)$. Thus, the quantum-mechanical Kohn–Sham energy, Equation (2), can be written as:(22)$$E^{QM}\left[\rho \right]=\;\overset{n}{\underset{i}{\sum }}\langle \psi_{i}|\hat{T}\psi_{i}\rangle -\;\overset{n}{\underset{i}{\sum }}\overset{QM}{\underset{A}{\sum }}\langle \psi_{i}\left|\frac{Z_{A}}{\left|r-A\right|}\right|\psi_{i}\rangle +\;\frac{1}{2}\overset{n}{\underset{i,j}{\sum }}\begin{array}{c} \langle \psi_{i}\psi_{i}∥\psi_{j}\psi_{j}\rangle + \end{array}E_{xc}\left[\rho \right]+E_{NN}$$

In the short-hand notation used here for the electron-repulsion integrals (ERIs), $\langle \psi_{i}\psi_{i}∥\psi_{j}\psi_{j}\rangle$, the symbol ∥ denotes the two-electron Coulomb operator $1/\left|r-r^{\prime }\right|$. It also separates functions of electron 1, on the left, from those of electron 2, on the right. In what follows, we will use an equivalent notation for ERIs over atomic orbitals and auxiliary functions. The upper sum limit n denotes the number of electrons in the system. To focus on the essentials, we will restrict the discussion to closed shell systems in which all occupied (occ) molecular orbitals (MOs) are doubly occupied. In the linear combination of atomic orbital (LCAO) approximation, the MOs in Equation (22) are expanded into atomic orbitals:(23)$$\psi_{i}\left(r\right)=\underset{\mu }{\sum }c_{\mu i}\mu \left(r\right)$$

In deMon2k μ(r) is expressed by atom-centered real (contracted) Cartesian or spherical-harmonic Gaussian type orbitals. With the MO coefficients at hand the closed-shell Kohn–Sham density can be expanded as:(24)$$\rho \left(r\right)=2\overset{occ}{\underset{i}{\sum }}\psi_{i}\left(r\right)\psi_{i}\left(r\right)=2\overset{occ}{\underset{i}{\sum }}\underset{\mu ,\nu }{\sum }c_{\mu i}c_{\nu i}\mu \left(r\right)\nu \left(r\right)=\underset{\mu ,\nu }{\sum }P_{\mu \nu }\mu \left(r\right)\nu \left(r\right)$$
The closed-shell density matrix elements are defined by:(25)$$P_{\mu \nu }=2\overset{occ}{\underset{i}{\sum }}c_{\mu i}c_{\nu i}$$
Thus, with the LCAO expansion, the closed-shell quantum mechanical Kohn-Sham energy can be written as:(26)$$E^{QM}\left[\rho \right]=\;\underset{\mu ,\nu }{\sum }P_{\mu \nu }H_{\mu \nu }\;+\;\frac{1}{2}\underset{\mu ,\nu }{\sum }\underset{\sigma ,\tau }{\sum }P_{\mu \nu }P_{\sigma \tau }\begin{array}{c} \langle \mu \nu ∥\sigma \tau \rangle + \end{array}E_{xc}\left[\rho \right]+E_{NN}$$

In Equation (26) the one electron terms, i.e., kinetic energy and nuclear attraction, are collected in the core Hamilton matrix elements $H_{\mu \nu }$. The computationally most demanding term in Equation (26) is the calculation of the four-center ERIs, 〈μν∥στ〉, which introduces a formal $N^{4}$ scaling, with N being the number of (contracted) basis functions. To overcome this computational bottleneck the variational fitting of the Coulomb potential, as pioneered by Dunlap and co-workers, ref. [17,45] is employed in deMon2k. The fitting is based on the minimization of the following error functional:(27)$$\varepsilon_{2}^{H}\left[\tilde{\rho }\left(r\right)\right]=\frac{1}{2}\iint \frac{\left[\rho \left(r\right)-\tilde{\rho }\left(r\right)\right]\left[\rho \left(r^{\prime }\right)-\tilde{\rho }\left(r^{\prime }\right)\right]}{\left|r-r^{\prime }\right|}drdr^{\prime }$$
In deMon2k the auxiliary density, $\tilde{\rho }\left(r\right)$, is expanded into atom-centered primitive Hermite Gaussian auxiliary functions indicate by a bar [46]: (28)$$\tilde{\rho }\left(r\right)=\underset{\bar{k}}{\sum }x_{\bar{k}}\bar{k}\left(r\right)$$

The expansion coefficients, $x_{\bar{k}}$, are called Coulomb fitting coefficients. Employing the expansion for ρ(r), Equation (24), and $\tilde{\rho }\left(r\right)$, Equation (28), expresses the fitting error functional, Equation (27), as:(29)$$\varepsilon_{2}^{H}\left[\tilde{\rho }\left(r\right)\right]=\;\frac{1}{2}\langle \rho ∥\rho \rangle -\langle \rho ∥\tilde{\rho }\rangle +\frac{1}{2}\langle \tilde{\rho }∥\tilde{\rho }\rangle =\;\frac{1}{2}\underset{\mu ,\nu }{\sum }\underset{\sigma ,\tau }{\sum }P_{\mu \nu }P_{\sigma \tau }\langle \mu \nu ∥\sigma \tau \rangle -\;\underset{\mu ,\nu }{\sum }\underset{\bar{k}}{\sum }P_{\mu \nu }\langle \mu \nu ∥\bar{k}\rangle x_{\bar{k}}+\;\frac{1}{2}\underset{\bar{k},\bar{l}}{\sum }x_{\bar{k}}x_{\bar{l}}\langle \bar{k}∥\bar{l}\rangle \geq 0$$
Because $\varepsilon_{2}^{H}\left[\tilde{\rho }\right]$ is positive semidefinite the following variational approximation for the two-electron Coulomb repulsion energy holds:(30)$$\frac{1}{2}\underset{\mu ,\nu }{\sum }\underset{\sigma ,\tau }{\sum }P_{\mu \nu }P_{\sigma \tau }\langle \mu \nu ∥\sigma \tau \rangle \geq \;\underset{\mu ,\nu }{\sum }\underset{\bar{k}}{\sum }P_{\mu \nu }\langle \mu \nu ∥\bar{k}\rangle x_{\bar{k}}-\;\frac{1}{2}\underset{\bar{k},\bar{l}}{\sum }x_{\bar{k}}x_{\bar{l}}\langle \bar{k}∥\bar{l}\rangle$$
The corresponding energy expression has the form:(31)$$E^{QM}\left[\rho \right]=\;\underset{\mu ,\nu }{\sum }P_{\mu \nu }H_{\mu \nu }+\;\underset{\mu ,\nu }{\sum }\underset{\bar{k}}{\sum }P_{\mu \nu }\begin{array}{c} \langle \mu \nu ∥\bar{k}\rangle \end{array}x_{\bar{k}}-\;\frac{1}{2}\underset{\bar{k},\bar{l}}{\sum }x_{\bar{k}}x_{\bar{l}}\langle \bar{k}∥\bar{l}\rangle +\;E_{xc}\left[\rho \right]+E_{NN}$$

As can be seen from Equation (31) the variational fitting of the Coulomb potential yields an energy expression free of four-center ERIs. Because the number of auxiliary functions is roughly 3 to 5 times the number of basis functions a cubic scaling energy expression is obtained. The variation of $E^{QM}\left[\rho \right]$, Equation (31), and $E^{QM*MM}\left[\rho \right]$, Equation (7) for QM/MM and Equation (16) for QM/MMpol, with respect to density matrix elements, keeping the Coulomb fitting coefficients constant, yields the Kohn–Sham matrix elements for QM/MM (Equation (33)) and QM/MMpol (Equation (34)):(32)$$K_{\mu \nu }=\;{\left(\frac{\partial E^{QM}\left[\rho \right]}{\partial P_{\mu \nu }}\right)}_{x}\;+{\left(\frac{\partial E^{QM*MM}\left[\rho \right]}{\partial P_{\mu \nu }}\right)}_{x}$$

(33)$$K_{\mu \nu }^{QM/MM}=H_{\mu \nu }+\;\underset{\bar{k}}{\sum }\langle \mu \nu ∥\bar{k}\rangle x_{\bar{k}}+\langle \mu |v_{xc}\left[\rho \right]|\nu \rangle -\;\overset{MM}{\underset{K}{\sum }}\langle \mu |\frac{q_{K}}{\left|r-R_{K}\right|}|\nu \rangle$$

(34)$$K_{\mu \nu }^{QM/MMpol}=H_{\mu \nu }+\;\underset{\bar{k}}{\sum }\langle \mu \nu ∥\bar{k}\rangle x_{\bar{k}}+\langle \mu |v_{xc}\left[\rho \right]|\nu \rangle -\;\overset{MM}{\underset{K}{\sum }}\left[\langle \mu |\frac{q_{K}}{\left|r-R_{K}\right|}|\nu \rangle -\langle \mu |\frac{\mu_{i}}{{\left|r_{i}-r\right|}^{3}}|\nu \rangle \right]$$

The exchange-correlation (XC) potential, $v_{xc}\left[\rho \right]$, is defined by the functional derivative of the exchange-correlation energy, $E_{xc}\left[\rho \right]$, with respect to the Kohn–Sham density. Alternatively, the variation of $E^{QM}\left[\rho \right]$ with respect to the Coulomb fitting coefficients, keeping the density matrix elements constant, yields a system of inhomogeneous equations for the determination of the Coulomb fitting coefficients:(35)$${\left(\frac{\partial E^{QM}\left[\rho \right]}{\partial x_{\bar{k}}}\right)}_{P}=\underset{\mu ,\nu }{\sum }P_{\mu \nu }\langle \mu \nu ∥\bar{k}\rangle -\;\underset{\bar{l}}{\sum }x_{\bar{l}}\langle \bar{l}∥\bar{k}\rangle \equiv 0$$

Casting these two variations, i.e., the energy minimization with respect to the MO coefficients under the constraints of MO orthonormality and the energy maximization with respect to the Coulomb fitting coefficients, into a Roothaan–Hall type SCF algorithm yields the four-center-ERI-free MinMax SCF [47] implemented in deMon2k. The computational efficiency of the QM/MM MinMax SCF is further improved by the double asymptotic expansion of the three-center ERIs [48] and the asymptotic expansion of the nuclear attraction type integrals in the QM/MM embedding term [49]. As a result, the numerical integration of the exchange-correlation energy and potential becomes the computationally most demanding task for the Kohn–Sham matrix construction in large scale QM/MM deMon2k calculations.

#### 2.2.2. Auxiliary Density Functional Theory (DFT)

In order to further improve computational efficiency, auxiliary density functional theory (ADFT) was developed and implemented in deMon2k [50]. In ADFT the linear scaling auxiliary density, $\tilde{\rho }\left(r\right)$, is used directly for the calculation of the exchange-correlation energy. As a result, the exchange-correlation energy expression in the QM energy, Equation (31), changes from $E_{xc}\left[\rho \right]$ to $E_{xc}\left[\tilde{\rho }\right]$. For the calculation of the ADFT Kohn–Sham matrix elements, we now need to derive $E_{xc}\left[\tilde{\rho }\right]$ with respect to density matrix elements:(36)$$\frac{\partial E_{xc}\left[\tilde{\rho }\right]}{\partial P_{\mu \nu }}=\;\int \frac{\delta E_{xc}\left[\tilde{\rho }\right]}{\delta \tilde{\rho }\left(r\right)}\frac{\partial \tilde{\rho }\left(r\right)}{\partial P_{\mu \nu }}dr$$
The functional derivative of the exchange-correlation energy defines the exchange-correlation potential,
(37)$$v_{xc}\left[\tilde{\rho };r\right]\;\equiv \;\frac{\delta E_{xc}\left[\tilde{\rho }\right]}{\delta \tilde{\rho }\left(r\right)},$$
now calculated with the auxiliary density. For the partial derivative of the auxiliary density with respect to the density matrix element [50,51]: (38)$$\frac{\partial \tilde{\rho }\left(r\right)}{\partial P_{\mu \nu }}\;=\;\underset{\bar{k}}{\sum }\frac{\partial x_{\bar{k}}}{P_{\mu \nu }}\bar{k}\left(r\right)=\;\underset{\bar{k},\bar{l}}{\sum }\langle \mu \nu ∥\bar{l}\rangle \langle \bar{l}∥{\bar{k}\rangle }^{-1}\bar{k}\left(r\right)$$
Back substitution of Equations (37) and (38) into (36) yields:(39)$$\frac{\partial E_{xc}\left[\tilde{\rho }\right]}{\partial P_{\mu \nu }}=\underset{\bar{k}}{\sum }\langle \mu \nu ∥\bar{k}\rangle z_{\bar{k}};\;z_{\bar{k}}\;\equiv \;\underset{\bar{l}}{\sum }\langle \bar{k}∥{\bar{l}\rangle }^{-1}\langle \bar{l}|v_{xc}\left[\tilde{\rho }\right]\rangle$$
The newly introduced exchange-correlation fitting coefficients $z_{\bar{k}}$ are spin dependent and calculated by numerical integration of the exchange-correlation potential. Because $\tilde{\rho }\left(r\right)$ is linear scaling the necessary grid work is greatly simplified. Furthermore, the ADFT QM/MM Kohn–Sham matrix elements depend only on the fitting coefficients $x_{\bar{k}}$ and $z_{\bar{k}}$:(40)$$K_{\mu \nu }=H_{\mu \nu }^{QM*MM}+\underset{\bar{k}}{\sum }\langle \mu \nu ∥\bar{k}\rangle \left(x_{\bar{k}}+z_{\bar{k}}\right)$$
By construction, the ADFT formalism so far introduced is restricted to “pure” density functionals, i.e., functionals that only depend on the auxiliary density and its derivatives. In order to extend ADFT to hybrid functionals that incorporate a certain fraction of exact (Fock) exchange the variational fitting of Fock exchange [52] is implemented in deMon2k. It is based on the maximization of the following negative semidefinite error functional:(41)$$\varepsilon_{2}^{EXX}\left[\tilde{\rho }\left(r,r\prime \right)\right]=-\overset{occ}{\underset{i,j}{\sum }}\iint \frac{\left[\rho_{ij}\left(r\right)-{\tilde{\rho }}_{ij}\left(r\right)\right]\left[\rho_{ij}\left(r^{\prime }\right)-{\tilde{\rho }}_{ij}\left(r^{\prime }\right)\right]}{\left|r-r^{\prime }\right|}drdr^{\prime }\;\leq 0$$
The Kohn–Sham and approximated MO product densities are given by:(42)$$\rho_{ij}\left(r\right)=\;\psi_{i}\left(r\right)\psi_{j}\left(r\right)$$
(43)$${\tilde{\rho }}_{ij}\left(r\right)=\;\underset{\bar{k}}{\sum }x_{\bar{k}ij}\bar{k}\left(r\right)$$
Note that the auxiliary functions for the expansion of the approximated density $\tilde{\rho }\left(r\right)$ and the approximated MO product densities ${\tilde{\rho }}_{ij}\left(r\right)$ are the same. Exploiting the variational nature of Equation (41) permits the calculation of the approximated exact exchange energy as:(44)$$E_{EXX}=\;-\overset{occ}{\underset{i,j}{\sum }}\underset{\bar{k},\bar{l}}{\sum }x_{\bar{k}ij}\langle \bar{k}∥\bar{l}\rangle x_{\bar{l}ij}\;=\;-\overset{occ}{\underset{i,j}{\sum }}\underset{\bar{k},\bar{l}}{\sum }\langle ij∥\bar{k}\rangle \langle \bar{k}∥{\bar{l}\rangle }^{-1}\langle \bar{l}∥ij\rangle$$
As Equation (44) shows, the variationally fitted exact exchange energy expression is free of four-center ERIs. However, the appearance of three-index fitting coefficients or ERIs leads to computationally unfavorable algorithms. To overcome this bottleneck localized MOs (LMOs) are used in the variational fitting of the Fock exchange. As a result, the auxiliary function sums in Equation (44) include only those functions relevant for the given i,j LMO combination. Thus, the generic hybrid ADFT QM energy expression has the form:(45)$$E^{QM}\left[\tilde{\rho }\right]=\;\underset{\mu ,\nu }{\sum }P_{\mu \nu }H_{\mu \nu }+\;\underset{\mu ,\nu }{\sum }\underset{\bar{k}}{\sum }P_{\mu \nu }\begin{array}{c} \langle \mu \nu ∥\bar{k}\rangle \end{array}x_{\bar{k}}-\;\frac{1}{2}\underset{\bar{k},\bar{l}}{\sum }x_{\bar{k}}x_{\bar{l}}\langle \bar{k}∥\bar{l}\rangle +\left(1-\alpha \right)E_{x}\left[\tilde{\rho }\right]-\alpha \;\overset{occ}{\underset{i,j}{\sum }}\overset{LMOs}{\underset{\bar{k},\bar{l}}{\sum }}\langle ij∥\bar{k}\rangle \langle \bar{k}∥{\bar{l}\rangle }^{-1}\langle \bar{l}∥ij\rangle E_{c}+\left[\tilde{\rho }\right]+E_{NN}$$
Here α denotes a generic hybrid functional mixing factor (not to be confused with atomic polarizabilities of atoms, e.g., $\alpha_{K}$). The corresponding hybrid ADFT QM/MM Kohn–Sham matrix elements are given by:(46)$$K_{\mu \nu }=H_{\mu \nu }^{QM*MM}+\underset{\bar{k}}{\sum }\langle \mu \nu ∥\bar{k}\rangle \left(x_{\bar{k}}+z_{\bar{k}}^{\prime }\right)-\;\alpha \overset{occ}{\underset{i}{\sum }}\overset{LMO}{\underset{\bar{k},\bar{l}}{\sum }}\langle \mu i∥\bar{k}\rangle \langle \bar{k}∥{\bar{l}\rangle }^{-1}\langle \bar{l}∥iv\rangle$$
With,
(47)$$z_{\bar{k}}^{\prime }\equiv \underset{\bar{l}}{\sum }\langle \bar{k}∥{\bar{l}\rangle }^{-1}\langle \bar{l}|\left(1-\alpha \right)v_{x}\left[\tilde{\rho }\right]+v_{c}\left[\tilde{\rho }\right]\rangle$$

In deMon2k, the QM/MM hybrid ADFT algorithm is implemented for global [53] and range-separated [54] hybrid functionals. For a more extended review of ADFT we refer interested readers to Reference [55]. Figure 2 compares SCF timings for the depicted QM system from a four-center ERI Kohn–Sham implementation [12] with corresponding three-center ERI ADFT timings for the generalized gradient approximation PBE [56], the global hybrid PBE0 [57], the long-range corrected PBE0 [58], the short-range corrected HSE06 [59] and the Coulomb-attenuating PBE0 functional [60]. All calculations were performed in parallel employing 12 Xeon X5675@3.07 GHz compute cores. As Figure 2 shows, the density fitting DFT (DF-DFT) and ADFT SCF timings are usually 1 to 2 orders of magnitude smaller than their four-center ERI counterparts. We also show QM/MM ADFT SCF timings (light blue bars) for the same system embedded in 100 MM water molecules. The comparison between QM ADFT and QM/MM ADFT SCF timings shows that the MM overhead is usually only a small fraction of the QM ADFT timings. For a more extended review of ADFT we refer the interested reader to Reference [55].

### 2.3. Available Methodologies

#### 2.3.1. Born–Oppenheimer Molecular Dynamics Simulations

Classical molecular dynamics (MD) is a well-known computer simulation technique where the time evolution of a set of interacting particles is followed by integrating their classical equations of motions. To start the computer simulation, the needed ingredients are a set of initial positions and velocities, by which the successive time evolution is in principle completely determined. With this information the computer calculates trajectories in a 6N-dimensional phase space, where N represents the number of atoms in the systems. Every point on the trajectory is a structure configuration. In this way, according to either an underlying statistical distribution function or to a statistical ensemble, a set of configurations is obtained. The physical quantities of interest are then represented by calculating averages over the available configuration sets. Therefore, expectation values of physical quantities can be obtained by calculating the arithmetic averages from the corresponding instantaneous values of these quantities along the simulated trajectories. In practice, simulation runs are always of finite length and one should be very careful to estimate if conformational sampling is sufficient. Although in a full description of the energetics and of the dynamics of a system, all parts involved, i.e., electronic and nuclear constituents should be treated quantum mechanically, physical and practical considerations, motivate calculations within the Born–Oppenheimer approximation, however. With this approximation a separation between the time scale of nuclear and electronic motions is introduced. Therefore, a Born–Oppenheimer molecular dynamics (BOMD) step consists of solving the static electronic structure problem, i.e., solving the stationary electronic Kohn–Sham equations, and the propagation of the nuclei via classical MD. In the BOMD step described above the computational bottleneck is represented by the solution of the Kohn–Sham equations. In deMon2k the computational demand for this task is considerably reduced by employing ADFT. It should be stressed that thanks to the variational nature of ADFT, reliable energy gradients, hence forces acting on nuclei, can be computed analytically, which is important to conserve energy in BOMD simulation in the microcanonical ensemble.

Figure 3 illustrates computational performances for a typical BOMD simulations at the QM/MM level. The simulated system consists in a uracil monophosphate molecule solvated in a water droplet. The QM region encompasses 185 atoms (i.e., the solute and its first hydration layer) while the MM region encompasses 10 578 atoms (3526 water molecules). We have used the DZVP-GGA basis set and the GEN-A2 (for H) and GEN-A2* (for C, N, O and P) auxiliary basis sets for a total of 1649 and 7193 atomic orbitals and auxiliary functions respectively. We have integrated XC contributions on the numerical grid with a tight accuracy (i.e., 10^−7^ Ha on the elements of the XC potential matrix). Simulations have been performed on Bull^®^ E5-2690V3@2.6 GHz processors on the OCCIGEN machine hosted at the CINES computer center (Montpellier, France). Each node has 64 Go of Random Memory Access (RAM) which enabled the use the MIXED scheme to store all near-field electron repulsion integrals in RAM [61]. It can be seen that noticeable gains can be obtained with up to 100 processors for this medium sized QM system. The most time consuming parts are XC contributions, density fitting operations, linear algebra (matrix diagonalization) and energy gradients calculations. We stress that evaluations of electron repulsion integrals are almost negligible when using MIXED ERIS scheme (ca. 3%). The right panel shows the energy evolution along QM/MM BOMD simulation in the microcanonical ensemble (after pre-equiibration at 300 K). We see that the average energy is stable after around 10 ps.

#### 2.3.2. Biasing Born–Oppenheimer Molecular Dynamics (BOMD) Trajectories

Because of the intrinsic cost of BOMD simulations it is very often necessary to bias trajectories in order to improve conformational sampling. This is for example the case if one wishes to estimate free energy profiles. One possibility is the use of restraints on distances (or difference of distances), angles or dihedrals that allows the construction of the potential of mean force for studying chemical reactions or conformational rearrangements [62,63]. Another approach is the metadynamics (MetaD) [64,65] tool available via the PLUMED 2.x package [26], which has been plugged into deMon2k [27]. For this article our original implementation for gas phase MetaD simulations has been extended to hybrid QM/MM and QM/Onsager MetaD simulations. During MetaD simulations, BOMD trajectories are biased by introducing a time-dependent external potential V(X,t) where ***R*** represents the atomic coordinates of the system. The dimensionality of V(X,t) is reduced by selecting a few collective variables (CVs), $s_{i}$:(48)$$V\left(X,t\right)=V\left(s_{1}\left(X\right),s_{2}\left(X\right),\ldots ,s_{n}\left(X\right),t\right)$$
and *V(**S**,t)* can then be expressed as a time-dependent sum of Gaussian functions:(49)$$V\left(S,t\right)=\int_{0}^{t}\omega exp\left(-\overset{n}{\underset{i=1}{\sum }}\frac{{\left[S_{i}\left(R\left(t\right)\right)-S_{i}\left(R\left(t^{\prime }\right)\right)\right]}^{2}}{2\sigma_{i}^{2}}\right)dt^{\prime }$$
where σ_i_ is the width of the Gaussian function associated with the ith CV, the summation runs up to the total number of CVs (n) and *S_i_(**R**)* is the value of the i^th^ CV that is expressed as a function of the atomic coordinates ***R***. Finally, ω is an energy rate defined by ω = W/τ_G_, where W and τ_G_ are the Gaussian height and the deposition stride, respectively. Biasing forces applied to atomic nuclei are obtained as derivatives of *V(**S**,t)* by application using the chain rule:(50)$$\frac{\partial V\left(X,t\right)}{\partial x_{k}}=\overset{n}{\underset{i=1}{\sum }}\frac{\partial V\left(X,t\right)}{\partial s_{i}}\frac{\partial s_{i}}{\partial x_{k}}$$

PLUMED 2.x offers a large variety of CVs and the user can also build new CVs as a function of simpler functions to adapt to any new chemical problem. Both on-the-fly biasing and post-processing analyses can be performed with PLUMED 2.x. To perform MetaD simulations with deMon2k one simply includes the keyword METADYNAMICS in the deMon2k input file and provides in a separate file (plumed.inpt) the input information needed for PLUMED 2.x calculations, such as definition of CVs, temperature and others. Detailed information about the PLUMED 2.x code and tutorial can be found in Ref [26]. An example application will be given in Part II.

#### 2.3.3. Electron Dynamics Simulations

deMon2k implements a module to conduct attosecond electron dynamics simulations under the frozen nuclei approximation via the so-called real-time time-dependent ADFT (RT-TD-ADFT) [24]. This methodology is useful to investigate the response of the electron cloud of a molecular system following a perturbation. The perturbation can be, for instance, a more or less strong electric field or the collision with a high energy charged massive particle. RT-TD-ADFT relies on the Runge and Gross theorem [66] and a Liouville–Von Neuman equation [67]. The time evolution of the density matrix is given by the commutator of P with the Kohn–Sham Hamiltonian H:(51)$$i\frac{\partial P^{\prime }\left(t\right)}{\partial t}=\;\left[H^{\prime }\left(t\right),\;P^{\prime }\left(t\right)\right]$$
the prime indicates that we have transformed the matrix into the basis of molecular orbitals [68,69]. This equation is solved numerically in deMon2k by using the second-order Magnus propagator [70]. Time is discretized into small time steps $\Delta t_{e}$ and the density matrix is evolved step-by-step as:(52)$$P^{\prime }\left(t_{i}+\Delta t_{e}\right)=e^{-iH^{\prime }\left(t_{i}+\frac{\Delta t_{e}}{2}\right).\Delta t_{e}}P^{\prime }\left(t_{i}\right)e^{-H^{\prime *}\left(t_{i}+\frac{\Delta t_{e}}{2}\right)*\Delta t_{e}}$$

$\Delta t_{e}$ is typically set to 1 to 10 attoseconds. As apparent from this expression, propagating the electron density from $t_{i}$ to $t_{i}+\Delta t_{e}$ requires the Kohn-Sham potential at $t_{i}+\frac{\Delta t_{e}}{2}$, which is a functional of the density at $t_{i}+\frac{\Delta t_{e}}{2}$, which is not yet known. To solve this problem both an iterative and a predictor-corrector [71] algorithm have been implemented [67,72]. The exponential of the matrix is evaluated by matrix diagonalization or by Taylor, Chebyshev or Baker–Campbell–Haussdorff expansions [73]. Interface to the ScalaPack library is possible for this computationally expensive task [74,75]. The coupling to polarizable and non-polarizable MM force fields has been described in Reference [24]. While interfacing with a non-polarizable force field is rather straightforward [76], the mutual polarization between the quantum and classical regions in the QM/MMpol framework makes propagation algorithms more involved [25,77,78]. We devised a stationary/non-stationary dual scheme [24] in which induced dipoles on MM atoms are fully relaxed at every electron propagation time steps. 

Overall our implementation of RT-TD-ADFT in deMon2k either in the gas phase or in the QM/MM and QM/MMpol framework is particularly efficient as it takes full advantage of the advanced algorithms described in Section 2.2. Figure 4 depicts a graph illustrating the performances of RT-TD-ADFT simulations. The system is the same as in Figure 3. We have relied on a DZVP-GGA/GEN-A2*(P, C, N)/GEN-A2(H, O) combination of atomic orbital and auxiliary basis sets, with grids of high accuracy to evaluate XC contributions and to extract atomic charges on-the-fly (every 5 as). An integration time step of 1 as has been used. We vary the size of the QM region by progressive inclusion of hydration layers as QM molecules. Our current implementation enables electron dynamics simulations for molecular systems including up one thousand of QM atoms in reasonable time. For the most demanding case of Figure 4 (878 atoms, 2988 electrons), it takes around 4 days to run 1 fs of simulation, and only 8 h for the systems comprised on 260 atoms. There is still room to further improve computational efficiency of RT-TD-ADFT with deMon2k and progresses are underway in our laboratories.

#### 2.3.4. Ehrenfest Molecular Dynamics Simulations

A natural extension of the electron dynamics simulations given by RT-TD-ADFT is its coupling with nuclear degrees of freedom. Ehrenfest MD [68,79] is a mean-field scheme belonging to the so-called Mixed Quantum-Classical (MQC) methodologies. MQC means that some degrees of freedom are treated quantum mechanically (in this case, electrons) while others are treated in the classical mechanics limit (in this case, nuclei). As in the BOMD module (see Section 2.3.1), nuclei are considered classical point particles, thus Newton’s equations of motion (EOM) apply for them. However, while in BOMD the nuclear and electron degrees of freedom are treated in a self-consistent manner, by relaxing the electrons for every new nuclear configuration invoking the so-called adiabatic approximation, EMD allows feedback between nuclear and electron motions.

The time-dependent (TD) solution of the coupled EOM starts by defining the general TD Schrödinger equations for both nuclei (53) and electrons (54) [79]:(53)$$i\hbar \frac{\partial \Omega \left(R,t\right)}{\partial t}=-\frac{\hbar^{2}}{2}\underset{b}{\sum }\frac{1}{M_{b}}\nabla_{R_{b}}^{2}\Omega \left(R,t\right)+\left\{\int \Psi^{*}\left(r,t\right){ℋ}_{r}\left(r,R\right)\Psi \left(r,t\right)dR\right\}\Omega \left(R,t\right)$$
(54)$$i\hbar \frac{\partial \Psi \left(r,t\right)}{\partial t}=-\frac{\hbar^{2}}{2}\underset{a}{\sum }\frac{1}{m_{a}}\nabla_{r_{a}}^{2}\Psi \left(r,t\right)+\left\{\int \Omega^{*}\left(R,t\right)V_{rR}\left(r,R\right)\Omega \left(R,t\right)dR\right\}\Psi \left(r,t\right)$$

The model introduces a feedback between the nuclear (Ω(R,t)) and electron (Ψ(r,t)) states in the quantities in curly brackets. Since such quantities are integrated, the potential given by one type of particle acting on the other type of particle is averaged. This will represent a disadvantage if the simulation traverses a region of possible states with very different energies since one may end up with wave functions that do not represent a physically realistic situation for the system under study. Thus, it is highly recommended to apply EMD in situations where electron states are relaxed in a time scale much shorter than the time scale for nuclei or ultrafast processes triggered by strong intense laser fields [80].

Considering the difference in the typical time scales for electrons (attoseconds) and nuclei (femtoseconds), it is reasonable to implement the EMD in such a way that the nuclear EOM are not necessarily solved for every electron step. In this way, molecular energy gradients can be calculated every few hundreds of electron steps, reducing the overall computational cost. An interpolation scheme borrowed from Reference [68] considers the solution of the nuclear EOM to obtain an initial and a final nuclear geometry.

### 2.4. How to Prepare a QM/MM Input for deMon2k?

In practice, a bottleneck when planning QM/MM simulations with a new program is the preparation of input files in appropriate formats. We explain in this section basic rules to set-up a QM/MM calculation with deMon2k. A program has been created to help users in this critical step. *PrepInpdeMon2k* enables one to switch from well-established packages (CHARMM [81], NAMD [82], Amber, Tinker [83], Tinker-HP [84]) for MM simulations to deMon2k for QM/MM simulations. The program reads topology/parameter files generated by AMBER (prmtop file), CHARMM (psf file) and coordinate file (pdb, AMBER coordinate or restart files, xyz files, CHARMM coordinate files) and generates a deMon2k template input file for a QM/MM calculation. During execution of *PrepInpdeMon2k* the user can specify QM/MM options and shape the simulated system according to the partition scheme illustrated in Figure 1. The need for link atoms is detected based on the list of QM atoms specified by the user.

The geometry needs to be supplied in the input file itself (deMon.inp) or in a separate geometry file (called deMon.geo) either in Cartesian coordinates or with a z-matrix. For large molecular systems (>5,000 atoms) using the deMon.geo file is recommended. Each line specifying an MM atom should contain, after the position specification, the force field (FF) atom type and possibly the atom connectivity. If the connectivity is not supplied, the program tries to determine it from distances with a criterion based on the covalent radius of the two atoms. To request addition of a link atom, a connection to the corresponding QM atom must be indicated on the line specifying the frontier MM atom. Based on the connectivity and on the selected force field the program builds the list of energy terms entering $E^{bonded}$ ($E^{bond}$,$E^{angle}$, $E^{torsion}$ and $E^{improper}$). However, the user has the possibility to modify afterward the list of MM terms by adding or deleting specific energy terms. It is also possible to modify the charge, the Lennard–Jones parameters or the atomic polarizabilities for specific atoms. This flexibility in the specification of MM energy terms may be extremely useful for non-standard residues necessitating specifically tuned FF.

Force field definition is contained in the external FFDS (force field dataset) file. We borrowed the parameter file format from the Tinker program [83]. FFDS files for AMBERf99 [29], Amberff02 [36], OPLS [23] and CHARMM22 [81] are available on the deMon2k website. It is also straightforward to integrate new atom types and FF terms in the FFDS file (an example will be given in part II).

## 3. Applications

In this section, we present six QM/MM applications to illustrate the kind of simulations that can be carried out with deMon2k. Applications involve both ground and excited electronic state processes. Tutorials with step-by-step descriptions of the simulations can be found on the deMon2k website [28]. They have been designed in a way to help users wishing to learn how to set up QM/MM calculations with deMon2k.

### 3.1. Organic Reactions ‘on Water’

Methods involved: saddle interpolation, intrinsic reaction coordinate (IRC) calculations and geometry optimization methods.

Experimental groups shown that several uni- and bi-molecular reactions are accelerated when performed in aqueous suspensions which are vigorously stirred. This procedure developed by Sharpless and co-workers is known as the “*on water*” protocol [75]. These type of reactions include different important classes of reactions known as cycloadditions, ene reactions, Claisen reactions and nucleophilic substitutions. Although before the work developed by Sharpless et al. there was already some evidence that many organic reactions are faster in water than in other organic solvents, it has been clarified only later that efficient reactions in aqueous organic chemistry do not require soluble reactants, as it was previously thought [76]. In the “on water” protocol reactions, the used reactants are initially floating on the surface of water, from where the “on water” designation comes from. The mixtures obtained are then stirred vigorously, dispersing the reactants as small droplets. The reaction products are often produced in a pure state under or on the water. These can be than isolated by phase separation of filtration techniques. In between several others fast reactions “on water”, it has been reported that the reaction of quadricyclane with dimethyl azodicarboxylate “on water” protocol amazingly runs to completion in only 10 min, much faster than with other procedures. However, the mechanism of why this reaction in water is so fast remains totally unclear [75,76]. Therefore, in the following example we will try to elucidate this mechanism of the reaction shown in Figure 5 by studying it in the gas phase and in water and compare the obtained potential energy barrier (Eb) in both cases.

Exchange-correlation effects are included within a hybrid functional proposed by Adamo and Barone [56,85] (PBE0). Local contributions were numerically integrated on a fine grid. The QM atoms were described with an all-electron double zeta valence plus polarization basis set optimized for GGA functionals (DZVP-GGA) [86]. For all calculations, the GEN-A2* auxiliary function set, which contains s, p, d, f and g auxiliary functions that are grouped together into sets with common exponents was employed. For the QM/MM level of theory, the QM part was as the methodology described above, whereas for the MM part, the empirical potential for 100 water molecules, with which the liquid environment was created, was a simple point-charge (SPC) model [87] employing the OPLS-AA Force Field [33].

Reactant and product structures were fully optimized without any constraints using a quasi-Newton optimization method [88,89]. The structure optimization was performed with the root mean square (RMSQ) gradient and the largest component of the gradient smaller than 10^−6^ Ha·bohr^−1^ and 1.5 × 10^−6^ Ha·bohr^−1^, respectively. In order to ensure that the obtained reactants and products structures are minima on the potential energy surface, the optimized structures were characterized by frequency analysis. For this purpose, the harmonic frequencies were obtained by diagonalizing the mass-weighted Cartesian force-constant matrix.

Once the product and reaction structures were obtained a transition state search was carried out. For the transition state search, a hierarchical transition-state-search algorithm was employed which combines the so-called double-ended Saddle interpolation method with the uphill trust region method [90]. First, double-ended saddle interpolations starting with the optimized reactant and product structures were employed to find appropriate start structures for the local transition state optimizations. In a second step, starting from these starting structures the transition states were optimized by an uphill-trust-region method. For the local quasi-Newton transition state optimizations, the start Hessian matrices were calculated. Once the transition state structures were optimized, as previously done with the product and reaction structures, these optimized structures were characterized by frequency analysis, too. Finally, to ensure that the obtained transition states indeed connect the reactants and products, the intrinsic reaction coordinates (IRCs) were calculated using a new reformulation of the Gonzalez and Schlegel algorithms. The saddle and IRC calculations have been performed only for the reaction in the gas phase.

In this work the chemical reaction of quadricyclane with dimethyl azodicarboxylate was studied with QM and QM/MM methodologies. In Table 2 the calculated potential energy barrier (E_b_) and reaction energies (Erxn) of the reaction of quadricyclane with dimethyl azodicarboxylate are reported in kcal·mol^−1^.

It can be observed from Table 1 that the calculated E_b_ in the liquid phase is lower by 7 kcal·mol^−1^ with respect to the potential energy barrier calculated for the same reaction in the gas phase. This result indeed indicates, in agreement with the experiment, that this kind of reaction occurs faster in liquid phase than gas phase. On the other hand, the calculated Erxn are in both cases very close to each other, as can be seen from Table 3. The energy profile is shown in Figure 6. 

In order to propose the mechanism of this reaction the transition states (TS) involved in the reaction were calculated and with the IRC calculations it was confirmed that these transition states were really connected with the reactants and products. Interestingly, in the gas phase the reactants (Quadricyclane and (*E*)-dimethyl azodicarboxylate) need to pass by two transition states and one intermediary structure (Quadricyclane and (*Z*)-dimethyl azodicarboxylate) to go to the product (1,2-diazetidine) as can be observed in Figure 6 (left) which illustrates this reaction profile.

The first transition state, TS1, corresponds only to the isomerization from (*E*)-dimethyl azodicarboxylate to (*Z*)-dimethyl azodicarboxylate. On the other hand, the second transition state, TS2, completes the reaction to go from the intermediary structure to the final product (Figure 6, left). In Table 3 are listed the first calculated harmonic frequencies of all structures involved in the reaction mechanism proposed for the chemical reaction of quadricyclane with dimethyl azodicarboxylate in gas phase and liquid phase.

To suggest a possible mechanism of this same reaction in the liquid phase “on water”, we considered all structures (product, reactant, transition state) found in the study of the reaction in the gas phase and inserted them inside a water ball formed from one hundred water molecules which was not previously optimized.

In order to avoid problems related to the orientation of the water molecules in the initial structures for which successive optimizations were performed, we took care that the water molecules forming the water bubble were always in the same orientation when inserting within them the structures obtained by the calculation performed in the gas phase. Finally, the optimization of all the resulting structures in the water environment were performed at the QM/MM level of theory. The results obtained are presented in Figure 6 (right) which illustrates the reaction profile of this reaction in the liquid phase. In order to have a better understanding of the orientation and of the structures of the molecules which have optimized within the water environment, in this figure these molecules are depicted at one side of the water ball (Figure 6, right). Important observations that can be mentioned here are the following:

(a) As can be seen from Figure 6 (right), the first transition state found in the gas phase disappears during the geometry optimization in water. Therefore, the chemical reaction in the liquid phase is characterized by only one transition state;

(b) The isomers (*E*)-dimethyl azodicarboxylate and (*Z*)-dimethyl azodicarboxylate do not survive in the liquid phase, the structure of dimethyl azodicarboxylate found is similar to the structure of TS1 found in the gas phase.

These results are very important because they demonstrate the influence of the water-molecule framework in the potential energy barrier and structures of the molecules that participate in this kind of chemical reaction. We need to bear in mind that the mechanism type reported in this work for the reaction of Figure 5 is just one possibility and other mechanisms could eventually exist. Therefore, this example opens an avenue to extend this study to other possible reaction mechanisms.

### 3.2. Umbrella Sampling for a Chemical Reaction

Methods involved: Umbrella sampling.

The study of the thermodynamics and kinetics of chemical reactions is often studied in the framework of free energy profile (also called potential of mean force, PMF) along a given reaction coordinate (RC) that connects the reactants and products of the reaction. One possibility to obtain such information is to perform umbrella sampling molecular dynamics (US-MD) simulations in which the system is driven along a predefined reaction coordinate X through the use of a biasing potential $V_{bias}$ added to the potential energy of the system:(55)$$V_{bias}\left(X\right)=\frac{1}{2}k_{bias}{\left(X-X_{0}\right)}^{2}$$
where $X_{0}$ is the target value that one wants to impose to the RC during the simulation, and $k_{bias}$ is the force constant of the harmonic biasing potential. Multiple simulations are run for different values of $X_{0}$ (windows) and the actual value of X is monitored during each of these simulations. Post-processing of the distribution of X in every windows yields the PMF along the reaction coordinate, for example using the weighted histogram analysis method (WHAM) [91,92]. The introduction of restraints and QM/MM features in deMon2k allows for the construction of PMF for chemical reactions with an explicit description of the environment. In order to illustrate this methodology, we consider a simple nucleophilic substitution following an S_N_2 mechanism:Nu^−^ + CH_3_–Cl → CH_3_–Nu + Cl^−^
In this application, we compare the nucleophilicity and leaving group ability of two similar Nu^−^ molecules: OH^−^ and SH^−^.

The system is built as follows. The AMBER suite of programs has been used to generate a chloromethane molecule system within a cubic box of water. A 1 ns-long force field-based MD simulation has been performed in (N,p,T) ensemble at 300K and 1 bar to equilibrate the solvent. The GAFF force field [29,93,94] is used to describe CH_3_Cl, and the TIP3P water model is chosen for the solvent. Periodic boundary conditions and the particle mesh Ewald method with a cutoff of 9 Å for the computation of non-bonded interactions are used. During the equilibration, one water molecule is kept close to CH_3_Cl, in a position adapted for the subsequent S_N_2 reaction. The final structure of the system is then used to prepare QM/MM simulations. We removed one proton from the water facing the carbon atom of CH_3_Cl to create the OH^−^ nucleophile (for the second S_N_ reaction, we also changed the oxygen atom by a sulfur atom). We also kept only the water molecules within a sphere of radius 30 Å around the solute. The QM part is composed of the OH^−^ (or SH^−^) anion and the CH_3_Cl molecule, while all the water molecules are considered at the MM level, using the TIP3P force field. The QM part is treated with the PBE functional and a DZVP-GGA basis set. Lennard–Jones potential parameters for the QM atoms are taken from the GAFF FF for CH_3_Cl, from TIP3P [95] FF for OH^−^, and from the AMBER FF of cysteine for SH^−^. For all the following MD simulations, a time step of 1fs is used and harmonic restraints (with a force constant of 10 kJ·mol^−1^·Å^−2^) on water-molecule atoms located further than 25 Å from the solutes are employed to prevent evaporation of the solvent. Moreover, we added a harmonic restraint to keep the alignment of the three atoms ((O or S), C and Cl) involved in the S_N_2 reaction.

A 500 fs MD simulation has been run at 10 K to prevent bad contacts due to the switch to the QM potential energy surface. The temperature of the system is gradually increased up to 300 K within 2.5 ps and a final 5 ps-long equilibration of the system is performed prior to the umbrella sampling simulations. The reaction coordinate that has been chosen to study the nucleophilic substitution is: $X=d_{CX}-d_{CCl}$, where $d_{CX}$ is the distance between the carbon atom and the oxygen or the sulfur atom and $d_{CCl}$ is the distance between the carbon and the chloride; 51 equally spaced windows have been used with targeted X values ranging from −5 to 5 Å. For each window we performed 1 ps MD simulations with a force constant of 250 kJ·mol^−1^·Å^−2^ for the biasing potential to bring X close to the target value. Production US-MD simulations were then performed over a period of 15 ps. In order to improve the sampling, we used a $k_{bias}$ force constant of 1000 kJ·mol^−1^·Å^−2^ for X values between −1.2 and 1.2 Å and added 14 intermediate windows within this range. A WHAM analysis of the simulations was performed using as input the RC values extracted every 10 fs over the last 10 ps for each window. Statistical uncertainties on the free energy values are obtained from a bootstrap error analysis using 50 samples. For the calculation of uncertainty, we have made the hypothesis that the correlation time for the reaction coordinate is 10 fs. This vale may be underestimated for some of the windows leading to error bars slightly underestimated as well. A precise evaluation of the uncertainty would require a more careful analysis of the autocorrelation function of the reaction coordinate and a longer sampling, which is beyond the scope of this methodological illustration. We used the 2.0.9.1 version of the freely available WHAM program of A. Grossfield to perform the WHAM analysis [96]. WHAM is freely available on Internet.

Figure 7 displays the free energy profiles obtained for the S_N_2 reactions with OH^−^ (in black) and SH^−^ (in red) respectively (note that positive values of the reaction coordinate correspond to [Nu^−^ + CH_3_Cl] configurations whereas negative values correspond to [Cl^−^ + CH_3_Nu]). For the sake of clarity, we have chosen not to represent error bars, which are smaller than 1 kJ·mol^−1^ for each curve over the full range of the reaction coordinate. The nucleophilic substitution of Cl^−^ by OH^−^ or SH^−^ is thermodynamically favored, with $\Delta_{r}G$ values roughly equal to −105 and −75 kJ·mol^−1^ respectively. The transition state for the reaction with OH^−^ is found for a reaction coordinate close to 0 and corresponds, as expected, to a structure where the 3 hydrogen atoms are within the same plane as the carbon atom. In the case of the reaction with SH^−^, $X^{\neq }$ is positive, around 0.35 Å, because of the bigger size of S with respect to O. For the forward reaction (Cl^−^ as leaving group), activation-free energies are very close for the two nucleophiles, roughly 65 kJ·mol^-1^. These results would point towards very similar nucleophilicity of OH^−^ and SH^−^. This is to be compared with gas-phase results where OH^−^ is found to be a better nucleophile than SH^−^ in similar S_N_2 reactions (see for example Gonzales et al. [97]). This may be explained by the stronger solvation in water of OH^−^ with respect to SH^−^. Looking at the backward reaction, one clearly sees that SH^−^ is a much better leaving group than OH^−^, which is in agreement with common knowledge in organic chemistry. These interpretations should however be taken with great care because many parameters could be improved in the simulations shown here (functionals, size of the water surrounding, addition of an implicit Onsager continuum, inclusion of few water molecules in the QM part, length of the sampling), that are only intended to exemplify the capabilities of deMon2k to build free energy profiles from QM/MM simulations.

### 3.3. Two-Dimensional Free Energy Surfaces

Methods involved: metadynamics, onsager continuum.

The performance of MetaD with QM/MM is illustrated by computing the free energy surface (FES) of the isomerization reactionconformational change of alanine dipeptide (dialanine) in a small water droplet of 233 water molecules. that has to be compared to the FES of dialanine in vacuum that displays only two main stable conformers (C7ax and C7eq in Figure 8B). The QM/MM model is shown in Figure 8A.

To this QM/MM model, the Onsager reaction field is applied to represent as a continuum (structureless) the water solvent around the explicit MM water molecules. Finally, to prevent evaporation of MM water molecules during the simulation, an Anderson containment sphere, defined by a radius of 17 Å, was applied. It bounces molecules back toward the center of the sphere while conserving energy and linear momentum. The BOMD simulations were performed at 300 K in the canonical ensemble using a Nosé-Hoover chain of 5 thermostats with frequencies of 400 cm^−1^. The integration time-step was set equal to 1 fs. The linear and the angular momenta were conserved with a threshold of 10^−8^ and, therefore, the rotational and translational degrees of freedom of the molecule were frozen. The MetaD simulations were performed using Gaussian functions defined by height and width of 0.4 kcal·mol^−1^ and 0.15 radian, respectively. They were added each 100 MD steps. The well-tempered formulation of MetaD with a bias factor of 10 was carried out for 370 ps trajectory length which corresponds to the deposition of 3700 Gaussian functions along the FES.

The two-dimensional FES of dialanine in water, presented in Figure 6 (C), displays a low free energy region at ϕ = −100 ± 30°. In this region, there are two distinct low-energy basins at (ϕ,ψ = (−100°, −10°) and (−100°, 120°), which correspond to the stable conformers α_R_ and β/C_7eq_, separated by 2–2.5 kcal/mol energy barrier. The α_R_ conformer appears only in the water phase owing its stabilization to the hydrogen bonds formed with the water molecules, described here only at the MM level. The stabilization of α_R_ conformer, demonstrates the good performance of the in-deMon2k electrostatic embedding QM/MM scheme (see Equations 6 and 7) in reproducing H-bond electrostatic character between the MM-water and QM-dialanine O, N and H atoms. Two high free-energy regions has ϕ coordinate in the intervals (−50°, 50°) and (100°, 180°) and is known to be constituted predominantly of less stable conformations. In the region around ϕ = 70 ± 20°, a second low-energy region is observed, although the corresponding free energies are higher compared to these in the ϕ = −100 ± 30° region. A quantitative sampling of the FES would necessitate a significantly longer simulation time. Nevertheless, although the present QM/MM simulations of alanine dipeptide in water are relatively short, only 370 ps, the landscape features of the FES resemble qualitatively those obtained previously with SCC-DFTB (Self-Consistent-Charge DFT Tight Binding) for 3.5 ns trajectory length [27] and other approaches [98,99].

### 3.4. Absorption Spectra of a Biological Chromophore

Method involved: linear response TD-ADFT, classical MD, QM/MM dynamics.

The simulation of the absorption and/or emission spectra of given chromophores is a crucial step to get photochemical information of the system under study. For instance, by comparing the simulated and experimental spectra we can assign the conformer or chemical form of the chromophore responsible for the absorption or emission. Moreover, accurate simulation of the absorption spectra by computation of the vertical transition energies lets us assign the electronic transitions corresponding to the different experimental absorption bands.

In this regard, it is advisable to study computationally a system that resembles the experimental conditions. That is, considering the interaction between the chromophore and the environment (solvent molecules or protein surroundings) is sometimes crucial to simulate spectra comparable to the experimental ones [100]. To illustrate the simulation of the absorption spectra in water solution, we have simulated the absorption spectra of the chromophore oxyluciferin (the light emitter in fireflies) in a solution of explicit water molecules (Figure 9). The absorption spectrum has been simulated by a gaussian convolution of a series of excitation energies computed at the QM/MM level for a statistical number of snapshots (sampling of water-oxyluciferin conformations) retrieved from a molecular dynamics simulation.

First, the system is built with the program tleap from the AMBER suite of programs by placing oxyluciferin in the center of an octahedral water box of 10 Å around the chromophore. The oxyluciferin parameters for the force field used at the classical level were obtained previously [100] and the water molecules are treated with the TIP3P model [95]. Afterwards, the system is equilibrated and minimized at the classical level with AMBER. At this point, we take the system and move to deMon2k for the heating step, by performing a progressive heating from 50 to 300 K. The preparation of the deMon input for the system built in AMBER is done with *PrepInpdeMon2k*. The water box is reduced to a sphere of 20 Å and the water molecules beyond 15 Å are subjected to harmonic constraints to keep the spherical shape of the atomic system.

After the heating, a classical MD simulation of 500 ps to sample diverse water-oxyluciferin conformations is performed; 100 snapshots from this MD simulation are extracted, and their vertical transitions computed at the QM/MM level using the local spin density approximation with standard local exchange functional and local correlation Vosko–Wilk–Nusair functional (VWN), the DZVP basis set and the GEN-A2 auxiliary basis set. In all cases, oxyluciferin is treated at the QM level and the water molecules at the MM level. Then, the energy values of only the excited singlet states are taken from each calculation and convoluted to give the absorption spectrum (Figure 10). To check the influence of the procedure followed to sample the ground state of oxyluciferin in water solution, we have also performed QM/MM molecular dynamics simulations of 200 ps following the classical MD one. In particular, two different QM/MM simulations were performed, one at a low-level (VWN functional, STO-3G as the basis set and GEN-A1 as the auxiliary set) of theory and the other one at a higher-level (PBE1 functional, DZVP as the basis set and GEN-A2 as the auxiliary set). 100 snapshots from each QM/MM simulation are extracted and the vertical electronic transition energies computed at the QM/MM level using the VWN functional, DZVP basis set and GEN-A2 auxiliary set as used for snapshots extracted from classical MD. The absorption spectra are simulated following the same procedure: convolution of Gaussian functions (Figure 10B). This way we can compare the simulated absorption spectra by sampling the water-oxyluciferin configurations following three different procedures: (i) classical MM MD simulation, (ii) low-level QM/MM MD simulation and (iii) high-level QM/MM MD simulation.

Analyzing the simulated and the experimental absorption spectra (Figure 10C), we conclude that the one obtained using classical MD is quite different from the experimental one, both in terms of shape and energy. Regarding those computed from snapshots extracted from QM/MM simulations, the general shape fits better with the experimental spectrum. In addition, the simulated spectra from the QM/MM simulations at low and high-level of theory are quite similar so, in this case we can perform the QM/MM simulation at a low-level of theory which implies a much lower computational cost. Although the maximum absorption wavelength is shifted a bit compared to the experimental one, the energy difference between the two absorption bands of the spectrum are close to experiment (0.82 eV experimentally, 0.88 eV in the QM/MM low-level procedure and 0.67 eV in the QM/MM high-level procedure). As a whole, the overall simulated spectrum (the two absorption bands corresponding to the excitation to two excited singlet states) is shifted in energy compared to the experiment. Agreement in the absolute values of the excitation energies (maximum absorption wavelengths) could be enhanced by modifying the level of theory used for the calculation of the electronic vertical transitions.

### 3.5. Electron Transfer Free-Energy Profile

Methods involved: BOMD, constrained DFT, polarizable QM/MMpol.

The objective of this tutorial is to evaluate the key parameters entering the rate expression of an electron-transfer (ET) reaction within the framework of Marcus Theory (MT) [101]. These parameters are the driving force (−ΔG°), the reorganization energy (λ) and the electronic coupling $H_{DA}$. We investigate the charge transfer between the two extreme tryptophan residues (W) of a pentapeptide, the sequence of which is WPPPW (W = Tryptophan, P = proline, see Figure 11). In the weak coupling regime, that is when $H_{DA}\ll \lambda$ and under the high temperature limit, the rate expression is expressed from the Fermi golden rule as:(56)$$k_{\mathrm{ET}}=\frac{2\pi }{\hbar }{\left|H_{\mathrm{DA}}\right|}^{2}\frac{1}{\sqrt{4{\pi \lambda k}_{B}T}}\exp\left[-\frac{{\left({\Delta G}^{0}+\lambda \right)}^{2}}{4{\lambda k}_{B}T}\right]$$
Marcus theory relies on the linear response approximation (LRA) which estimates ΔG° and λ from microscopic simulations by application of the two following equations [102,103]:(57)$$\Delta G^{0}=\frac{{\langle \Delta E\rangle }_{1}+{\langle \Delta E\rangle }_{2}}{2}$$
(58)$$\lambda =\frac{{\langle \Delta E\rangle }_{1}-{\langle \Delta E\rangle }_{2}}{2}=\lambda^{St}$$
where $\Delta E=E_{2}-E_{1}$ is the vertical potential energy gap between the final and initial charge transfer states. ${\langle \Delta E\rangle }_{x}$ denotes the thermal average of ΔE for the system in electronic state x (1 or 2). $\lambda^{St}$ is known as the Stokes reorganization energy. Using ΔE as the global reaction coordinate [104] and defining the probability distribution $p_{x}\left(\varepsilon \right)$ to be the probability to have ε=ΔE, the free energy function for the system in each redox states is obtained by the Landau formula: $g_{x}\left(\varepsilon \right)=-\beta ln\left(p_{x}\left(\varepsilon \right)\right)+g_{x}^{0}$ ($\beta =1/k_{B}T$). Assuming the LRA holds means that $p_{x}$ have Gaussian shapes, or equivalently that $g_{x}$ are parabolic.

For each specific ET system, microscopic simulations can be used to (1) test the validity of the different hypotheses underlying Marcus theory for this specific ET reaction (2) provide estimates for each of the quantities entering the theory. A key point is that the theory relies on the notion of “*ad hoc*” diabatic charge transfer state so that adequate QM methods must be used. deMon2k implements a hybrid QM/MM scheme relying on constrained DFT [105,106] (cDFT) to define the charge transfer states underlying MT [107,108]. cDFT is well adapted for net electron transfers because the excess charge (or excess electron) is localized on the electron donor (D) or on the acceptor (A).

To model the ET reaction depicted in Figure 11 we first carry out BOMD simulations at the hybrid cDFT/MM simulations in the two redox states. For the FF we used with the non-polarizable Amberff99 FF [29] and the TIP3P model for water. To reduce computational cost we employ a minimal STO-3G basis set and the GEN-A2 auxiliary basis set. The QM system is composed of the two extreme W residues (placing the QM/MM frontiers at the Cα-Cβ bonds). Hydrogen link atoms are used to saturate the valence of Cβ atoms. The MM system consists of a water sphere of 33 Å. A spherical Onsager solvation model is used for remote interactions with a dielectric constant of 80. During the simulation the atoms situated in the most remote 3 Å-width water shell are subjected to harmonic restraints on their initial positions to prevent water diffusion into the continuum. After progressive heating from 10 to 300 K during 2.8 ps, the system is simulated on each of the potential energy surfaces (PES) corresponding to the two charge transfer states for 10 ps using a cDFT/MM methodology. The trajectories are then reprocessed to evaluate ΔE and $H_{DA}$ every 20 fs. For the reprocessing calculations we have switched to the DZVP-GGA basis set and used either a non-polarizable or a polarizable force field.

$\Delta G^{0}$ and $\lambda^{St}$ amount to −0.25 and −0.18 eV respectively with Amberff99/TIP3P FF and to 1.90 and 1.62 eV with the Amberff02/POL3 FF. Note the decrease of $\lambda^{St}$ when the electronic induction effect is included in the FF [109]. Note that a more pronounced decrease of $\lambda^{St}$ upon inclusion of induction would have been obtained if we had also run the BOMD with MMpol. The electronic coupling $H_{DA}$ between the two charge transfer states should be computed in principle only for configurations of near-degeneracy of the two states. It is possible to bias the BOMD trajectories to sample only regions of the phase space associated to ΔE~0 by imposing a harmonic potential on the vertical energy gap [108]. For space reasons this option is not illustrated in this tutorial, however. The average electronic coupling amounts to 86 and 82 meV along MD simulations of the initial and final states with standard deviations of 43 and 92 meV. The so-called coherence parameter [110] defined as $C_{coh}=\langle H_{DA}\rangle {}^{2}/\langle H_{DA}^{2}\rangle$ takes values of 0.006 and 0.003, very close to zero, which indicates that the peptide conformational dynamics strongly affects ET. Other applications of cDFT to model ET processes with deMon2k can be found in References [111,112,113].

### 3.6. Non-Adiabatic Chemistry Induced by Ionizing Radiation

Method involved: real-time TD-ADFT, Ehrenfest MD.

This tutorial illustrates the electron dynamics module implemented in deMon2k [24]. Two types of simulations are available. The first one is purely RT-TD-ADFT under the frozen-nuclear-position approximation. The second one is mean-field Ehrenfest MD which couples electron dynamics to nuclear dynamics. The former method is well suited to simulate electronic processes taking place on ultrashort time scales, from attoseconds up to a few femtoseconds, before atomic nuclei have time to respond. Both methods are compatible with QM/MM with either non-polarizable or polarizable FF. We consider in this tutorial the collision of a high-kinetic-energy proton with a peptide. Fast protons interact primarily with the electron cloud by Coulomb interactions inducing ionization of the molecule on the attosecond timescale, and eventually fragmentation of the molecule on longer, femtosecond, time scales. We simulate a five amino acid peptide (tyrosine-glycine-glycine-phenylalanine-methionine) solvated in a sphere of radius 29 Å (Figure 11). To allow fast simulations for this tutorial we restrict the QM region to the terminal methionine side chain. A link atom saturates the valence of the Cβ atom. The projectile moves along the x direction as shown in Figure 12; it is initially placed 50 Å away from the QM region. We use the aug-cc-pVTZ basis set [114] on sulfur and hydrogen atoms and a very diffuse basis set from Reference [115] for carbons. To remove emitted electrons from the system a complex absorbing potential is placed 10 Å away from the QM region [25,116]. We first conducted purely RT-TD-ADFT to simulate energy deposition and ionization. In this simulation the proton has no possibility to deviate from its trajectory (inelastic collision). The simulations have been run for 5 fs with a time step of 1 as. Hirshfeld atomic charges were calculated every 20 as using the auxiliary density [117]. As seen on the graph of the right panel of Figure 12 almost 100 eV are deposited within the electron cloud, which is clearly a very high value. The electron cloud undergoes very fast dynamics as testified by the fluctuations of the atomic charges (Figure 12, right bottom).

The energy deposited upon collision is eventually dissipated within nuclear degrees of freedom. To explore this eventuality we have carried out Ehrenfest MD. The system was first pre-equilibrated during 5 ps BOMD simulation in the canonical ensemble before switching to simulation of elastic collision by fast protons. Ehrenfest MD have been propagated for 20 fs with a nuclear propagation time step of 10 as and an electronic propagation time step of a third of an attosecond. We find that within 20 fs the methionine side chain is fragmented into a CH_3_ radical and the remaining side chain (Figure 13). The length of the hydrogen bond between sulfur and surrounding water increases too as a consequence of ionization of the side chain. Overall on this time scale, the energy deposited upon collision doesn’t have time to spread out within surrounding nuclear modes but before covalent bond breaking has time to take place. 

## 4. Conclusions

In this article we have reviewed the methodologies implemented in deMon2k for conducting QM/MM simulations taking advantage of the auxiliary DFT framework. Available methodologies encompass ground state Born–Oppenheimer MD, non-adiabatic MD simulations within the mean-field Ehrenfest scheme and attosecond electron dynamics simulations. deMon2k undergoes continuous improvements and work is underway in our laboratories to further enhance its QM/MM methods. For example more efficient algorithms for geometry optimizations, MD propagation (e.g., time-reversible BOMD of Niklasson et al. [118]) would be desirable. Our groups also work toward the implementation of other force fields like CHARMM36 [119] or polarizable force field. Interested users or developers can obtain the source code from the deMon2k website (http://www.demon-software.com).

Six examples of applications have been reported illustrating various kinds of simulations, reaction mechanism searches on potential energy surfaces, simulations for building 1D and 2D free-energy surfaces, simulations of absorption spectra or direct simulations of radiation-induced physical-chemical processes. To keep simple tutorial we have focused on small systems embedded in water. There is no obstacle to address QM/MM simulation of more complex molecular systems like proteins or DNA with the current implementation. Interested users can download tutorials on the deMon2k website to reproduce these simulations.

## Figures and Tables

**Figure 1 molecules-24-01653-f001:**
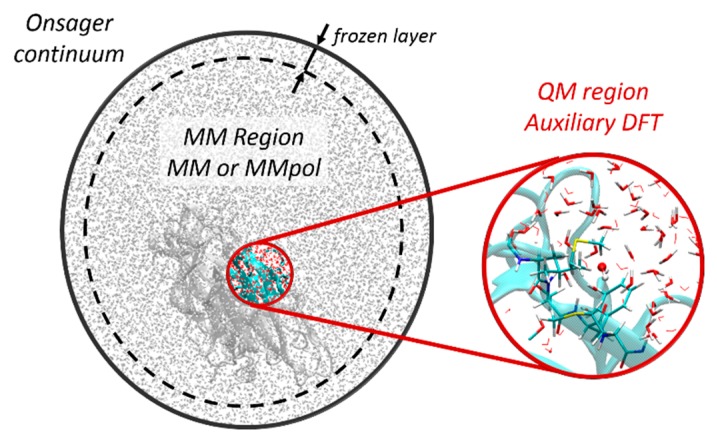
A recommended partition for Quantum Mechanical/Molecular Mechanical (QM/MM)/Onsager simulations with deMon2k.

**Figure 2 molecules-24-01653-f002:**
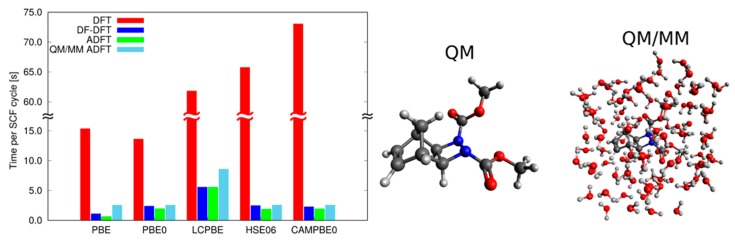
Comparison of Kohn–Sham Density Functional Theory (DFT) self-consistent field (SCF) timings (red) with corresponding auxiliary density functional theory (ADFT) QM (green) and QM/MM (light blue) SCF timings.

**Figure 3 molecules-24-01653-f003:**
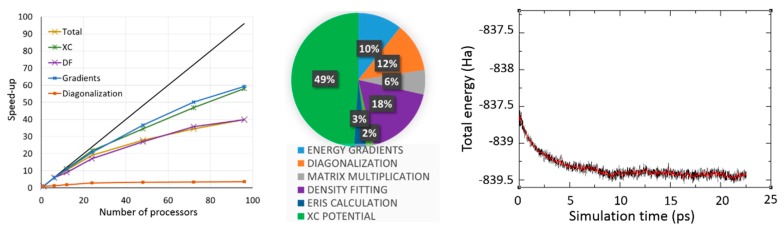
Computational performances for QM/MM Born–Oppenheimer molecular dynamics (BOMD) simulation. Left: speed-up as a function of number of processors. Only tasks representing more than 15% of total execution time are depicted. Middle: pie chart illustrating the most computationally demanding task (taking the job run on 48 processors). Right: energy conservation for in the microcanonical simulations.

**Figure 4 molecules-24-01653-f004:**
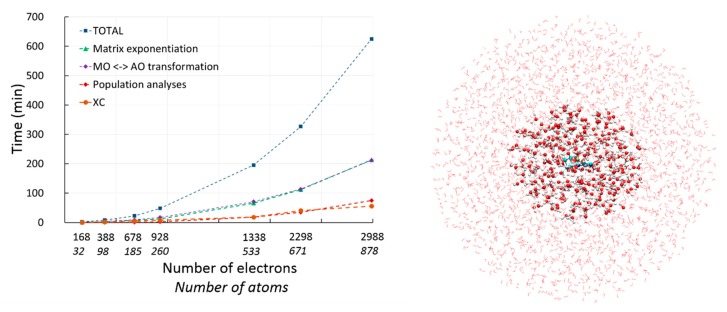
Computational performances for QM/MM RT-TD-ADFT simulation. Left: scalability of code with size QM region. Right: Simulated system with QM and MM atoms represented with balls-and-sticks and lines respectively. The picture represents the QM/MM calculation with the largest QM systems (878 atoms).

**Figure 5 molecules-24-01653-f005:**
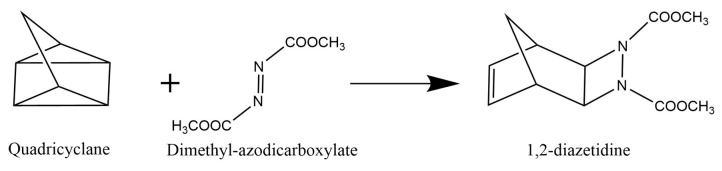
Reaction of quadricyclane with dimethyl azodicarboxylate to yield 1,2-diazetidine.

**Figure 6 molecules-24-01653-f006:**
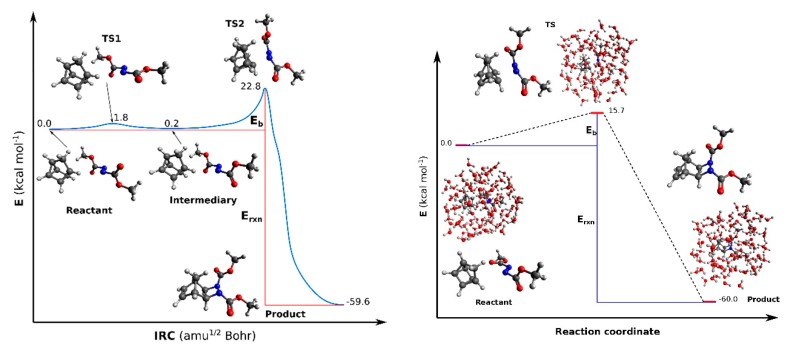
Reaction profile of quadricyclane with dimethyl azodicarboxylate in the gas and liquid phase (respectively, left and right panels).

**Figure 7 molecules-24-01653-f007:**
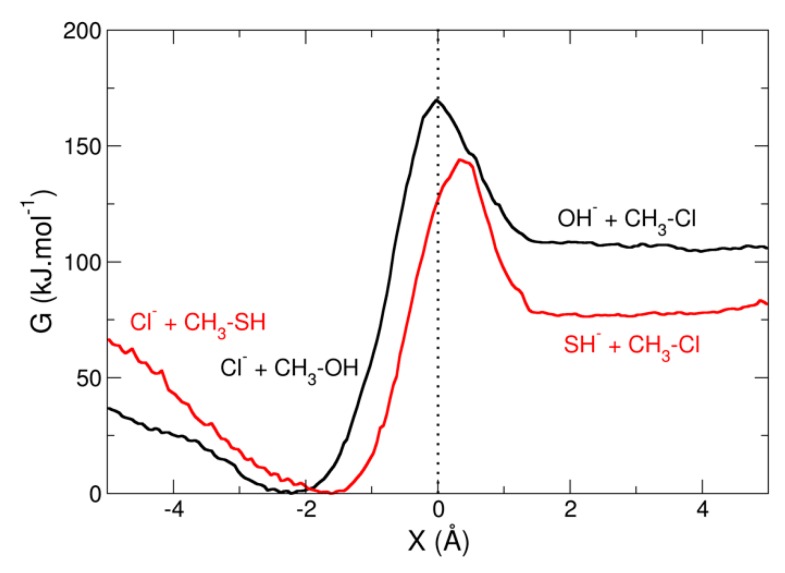
Free energy profiles for S_N_2 reactions between CH_3_Cl and OH^−^ (in black) or SH^−^ (in red).

**Figure 8 molecules-24-01653-f008:**
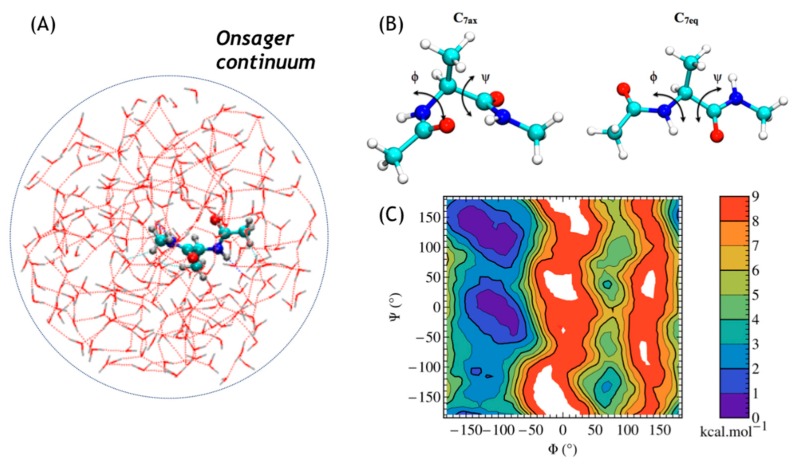
(**A**) A snapshot of the QM dialanine (ball and stick) and MM water (lines) model; (**B**) the two most stable conformations of alanine dipeptide in vacuum, C7_ax_ and C7_eq_, with the backbone dihedral angles φ and ψ and (**C**) two-dimensional free energy surface (FES), projected on φ and ψ angles. The atom color is the following: C-cyan; N-blue; O-red and H- grey.

**Figure 9 molecules-24-01653-f009:**
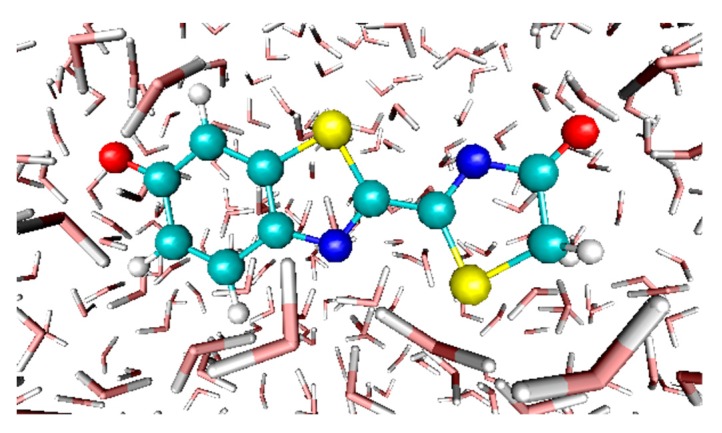
Representation of the system under study: oxyluciferin (CPK drawing method) and the water molecules mimicking water solution.

**Figure 10 molecules-24-01653-f010:**
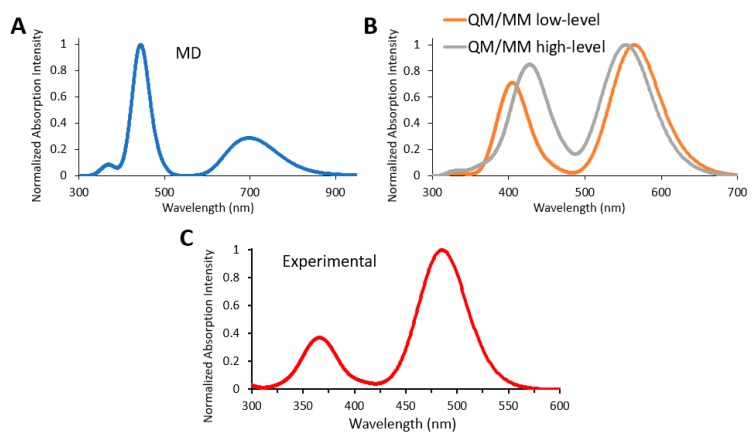
Simulated absorption spectra taking the snapshots from (**A**) the classical MD simulation and (**B**) the QM/MM simulations both at the low and high levels of theory. (**C**) Experimental absorption spectrum of oxyluciferin.

**Figure 11 molecules-24-01653-f011:**
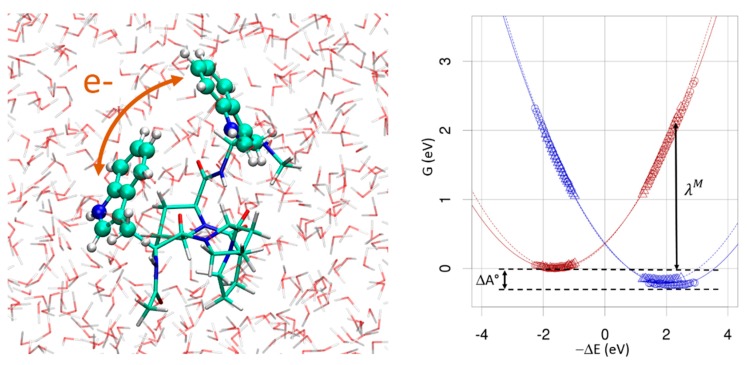
Constrained DFT/MM simulation of charge transfer between tryptophane residues. Left: zoom on the peptide. The atoms with ball-and-stick representation are treated at the QM level. All other atoms are treated with force field (FF). Right: Marcus theory free energy profiles taking the vertical energy gap as reaction coordinate. Circles and triangles correspond to simulation data obtained from non-polarizable and polarizable FF respectively. The lines correspond to parabolic regressions on the simulation data (with induction in dashed lines, without induction in plain lines). The points at the bottom of each free energy curves are directly coming from the simulation data while those are obtained by applying ΔG(ε)=ε which derives from the ergodic hypothesis (see Reference [102] for demonstration).

**Figure 12 molecules-24-01653-f012:**
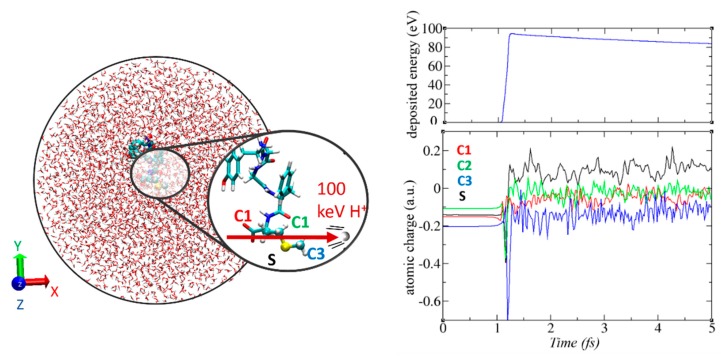
Collision of terminal methionine residue by a 100 keV proton. Left: QM/MM set-up. Only methionine residue is treated as QM (balls and sticks), the rest of the system pertains to the MM region. Right: Variation of QM/MM energy of the system and charges of the carbon and sulfur atoms region.

**Figure 13 molecules-24-01653-f013:**
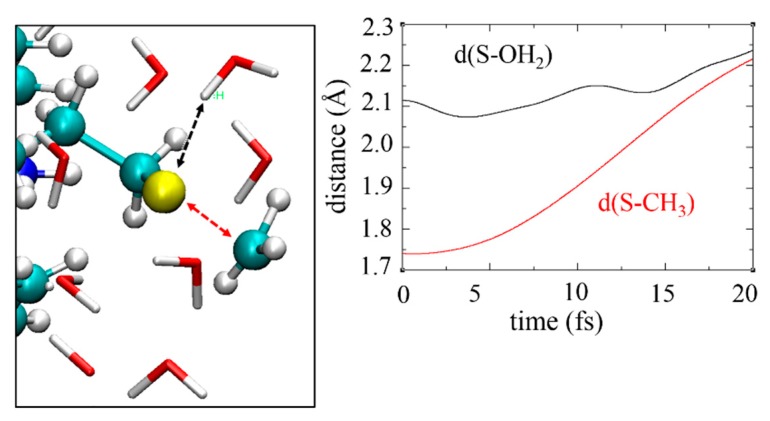
Fragmentation of methionine side chain after collision by high energy proton. Left: snapshot at 20 fs. Right: evolution of two distances involving S atom.

**Table 1 molecules-24-01653-t001:** New features of deMon2k for quantum mechanical/molecular mechanical (QM/MM) simulations with respect to 2015 review paper [19].

Methods	References
Polarizable embedding	[24]
Attosecond electron dynamics	[24,25]
Ehrenfest non-adiabatic electron-nuclear dynamics	-
Link atoms	this work
Continuum solvation model for long range interactions	this work
Tutorials	this work
Geometrical restraints	this work
Metadynamics via interface to plumed library [26]	[27], this work

**Table 2 molecules-24-01653-t002:** Comparison of potential energy barrier (Eb) and reaction energy (Erxn) in gas phase and liquid. All values are in kcal·mol^−1^.

Gas Phase (QM)		Liquid Phase (QM/MM)	
E_b_	E_rxn_	E_b_	E_rxn_
22.8	−59.6	15.7	−60.0

**Table 3 molecules-24-01653-t003:** First harmonic frequency (in cm^−1^) of all structures reported in the mechanism of the reaction in gas phase and liquid phase.

Gas Phase (QM)					Liquid Phase (QM/MM)		
Reactant	TS1	Intermediary	TS2	Product	Reactant	TS	Product
9.2	61.0i	8.3	504.4i	58.8	15.2	372.8i	13.9

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
