# Peer review of "Molecular Simulations with in-deMon2k QM/MM, a Tutorial-Review"

_molecules, 2019, doi:10.3390/molecules24091653_

Reviewer 1 Report

The paper by de la Lande and coworkers describes recent advancements in QM/MM molecular simulations with deMon2K. The paper is nicely presented and gives basic theory and applications, covering the main features of the implementation discussed in this manuscript. In general, I think that this paper is a good contribution and that deserved publication in Molecules. There are, however, a number of issues that could be considered by the authors in order to improve the manuscript:

1.     Section 2.1.3 describes both the implementation of a continuum model and also the Link atom method. I found this confusing and they should be separated in two different sections. In addition, the authors should state if other treatments of long-range interactions (Ewald summation) are included in the current implementation. It would be also convenient to recall that the continuum model implementation discussed in the manuscript requires the neutrality of the simulated sphere.

2.     In general I missed details about the computational efficiency of the implementation. Only the use of Auxiliary DFT is discussed in terms of efficiency. It would be of interest for the reader to provide information about the computational time required in each of the applications presented.

3.     In addition, I also missed the comparison of the results presented in this work with previous results or even experimental data. Some of the discussed systems have been analyzed in detail and used as a test in many implementations (see section 3.3). It would be of interest for the user of deMon2K to compare the results obtained with other QM/MM programs.

4.     The use of the term ‘activation energy’ is confusing. The activation energy is an experimental parameter that can be obtained from the analysis of the temperature dependence of the rate constant. It would be much more convenient to use another terminology in section 31., such as ‘potential energy barrier’ or ‘classical potential energy barrier’.

5.     It is difficult to admit that the statistical error obtained in the free energy profiles presented in section 3.2 is smaller than 1kJ/mol. Does this error refer to the free energy difference between products and reactants?

In this section it would be of interest too to discuss the effect of long range interactions in the free energy profile.

6.     The authors could add a comment about the origin of the large difference observed between the results obtained in the absortion spectrum of oxyluciferim when classical or QM/MM simulations are employed (figures 8 A and B).

Author Response

Reviewer 1:

The paper by de la Lande and coworkers describes recent advancements in QM/MM molecular simulations with deMon2K. The paper is nicely presented and gives basic theory and applications, covering the main features of the implementation discussed in this manuscript. In general, I think that this paper is a good contribution and that deserved publication in Molecules. There are, however, a number of issues that could be considered by the authors in order to improve the manuscript:

"1.     Section 2.1.3 describes both the implementation of a continuum model and also the Link atom method. I found this confusing and they should be separated in two different sections. In addition, the authors should state if other treatments of long-range interactions (Ewald summation) are included in the current implementation. It would be also convenient to recall that the continuum model implementation discussed in the manuscript requires the neutrality of the simulated sphere."

We have split Section 2.1.3 into two sections. Section 2.1.3 describes long range electrostatics while Section 2.1.4 describes the link atoms implementation.

In section 2.1.3 we now state that periodic boundary conditions are not available in the current implementation and that for charged simulation sphere a Born model is implemented to evaluate solvation energy.

"2.     In general I missed details about the computational efficiency of the implementation. Only the use of Auxiliary DFT is discussed in terms of efficiency. It would be of interest for the reader to provide information about the computational time required in each of the applications presented."

The efficiency is Auxiliary DFT is effectively emphasized in the code. Following recommendations of Reviewer 2 (see below), we now provide in Part I (section 2.3) information about computational time required in typical Born-Oppenheimer molecular dynamics simulations and in electron dynamics simulations. 

"3.     In addition, I also missed the comparison of the results presented in this work with previous results or even experimental data. Some of the discussed systems have been analyzed in detail and used as a test in many implementations (see section 3.3). It would be of interest for the user of deMon2K to compare the results obtained with other QM/MM programs."

Although the reviewer's suggestion is valuable, we wish to emphasize that our objective in thus article  is not to compare different QM/MM codes but to provide simple examples with the aim of helping potential users to learn how to use deMon2k for QM/MM. This is from our point of view clearly explained in Introduction. We thus haven't modified our text with respect to this comment.

"4.     The use of the term ‘activation energy’ is confusing. The activation energy is an experimental parameter that can be obtained from the analysis of the temperature dependence of the rate constant. It would be much more convenient to use another terminology in section 31., such as ‘potential energy barrier’ or ‘classical potential energy barrier’".

The Reviewer is right. We have corrected the text accordingly in section 3.1.

"5.     It is difficult to admit that the statistical error obtained in the free energy profiles presented in section 3.2 is smaller than 1kJ/mol. Does this error refer to the free energy difference between products and reactants?"

The value of 1kJ/mol is the standard deviation obtained through a bootstrap analysis of the computation of the free energy profile. It is for all the points where the free energy is calculated, not only for products and reactants. We have made the hypothesis that the correlation time between successive values of the reaction coordinate is 10fs, which may be under-evaluated for some of the windows. We have added a sentence to clarify this in the text. A better evaluation of the correlation time would be needed to better define statistical uncertainty, but it is beyond the scope of this methodological paper.

"In this section it would be of interest too to discuss the effect of long range interactions in the free energy profile."

We agree with the reviewer that treatment of long range interactions is quite crude in this particular application. For example a surrounding Onsager could have been added, or we could have used a larger solvation sphere. It is probable that the quantitative results would have been different. As mentioned in the last paragraph of section II.2, many parameters of the calculation may be improved to get more reliable results (we have added to the list suggestions about running simulations with a larger water sphere or with a surrounding continuum Onsager model). However, we have voluntarily made the choice to keep the simulations as simple as possible to be easily run as a tutorial. 

"6.     The authors could add a comment about the origin of the large difference observed between the results obtained in the absortion spectrum of oxyluciferim when classical or QM/MM simulations are employed (figures 8 A and B)."

We agree with the referee that a comment should be added in the main text to explain the different spectra simulated when classical or QM/MM simulations are used. For this aim, we have added a sentences in the main text: “The low agreement obtained between the experimental spectrum and the one simulated using classical MD could be explained by a poor sampling due to the quite short MD simulation performed.”

Reviewer 2 Report

The authors presented a review of the features included in the deMon2k software for Computational Chemistry. This software could be of major interest to chemists especially those who work in the topic of excited states. Thus, the reviewer considers that the present paper is suitable for publication in Molecules journal. However, the reviewer suggests several comments which must be addressed prior to publication, those are:

*Do the authors use a switching function between QM and MM regions in Eqs. 7 and 16 to avoid discontinuities? These could be appreciated in an NVE simulation as an energy drift. 

*A plot for energy conservation would be instructive to the reader in order to see how well ADFT method conserves energy in the context of QM/MM methodology.

*Although ADFT reduces the computational time per SCF step, it doesn't reduce the number of steps. An approach such as Time Reversible BOMD of Niklasson, et. al could be useful here.

*A plot showing the scalability of the code for a medium-size system is desired, for instance, time (or speedup) vs. nr. of MPI ranks.

*Is the present version a pure MPI implementation or is that a hybrid MPI-OpenMP one?

*Something that is not clearly mentioned is what differentiates the present software from other existing packages for performing QM/MM simulations? This could be mentioned in the introduction and also in the conclusion sections.

*Although QM/MM simulations for watery environments are suitable for testing the performance of the software, a catalytic reaction in an enzymic environment would be more interesting for realistic applications.

*In the manuscript, it is mentioned that "Tutorials with step-by-step descriptions of the simulations can be found on the deMon2k website". The reviewer couldn't find that set of tutorials. Thus, we recommend the authors to provide a direct link to the tutorials they referred to.

*CHARMM22 topology/parameters set is obsolete, it is recommended to support instead CHARMM36 instead.

*A more standard SN2 reaction of Cl+CH3Cl which has been reported previously in the literature would be instructive at this point. In this way, the reader could see how the methods implemented in the present work perform.

*A section with all novel features of the software in the form of a list of a table is desired.

In addition to the comments mentioned above, the reviewer detected several "not found" references throughout the text. 

Author Response

The authors presented a review of the features included in the deMon2k software for Computational Chemistry. This software could be of major interest to chemists especially those who work in the topic of excited states. Thus, the reviewer considers that the present paper is suitable for publication in Molecules journal. However, the reviewer suggests several comments which must be addressed prior to publication, those are:

"*Do the authors use a switching function between QM and MM regions in Eqs. 7 and 16 to avoid discontinuities? These could be appreciated in an NVE simulation as an energy drift."

In the present implementation no switching function between QM and MM are used between the QM and MM regions. All energy terms are included in the calculations. Of course we are aware of more elaborated schemes and work is underway in our laboratories to further improve our methodology.

"*A plot for energy conservation would be instructive to the reader in order to see how well ADFT method conserves energy in the context of QM/MM methodology."

Conservation of energy is important indeed. We have added a graphic showing energy conservation in QM/MM simulations with ADFT (section 2.3.1, and Figure 3).

"*Although ADFT reduces the computational time per SCF step, it doesn't reduce the number of steps. An approach such as Time Reversible BOMD of Niklasson, et. al could be useful here. "

We thank the reviewer for this interesting comment. We have added a citation to the paper   of Time Reversible BOMD of Niklasson, et. al in conclusion with appropriate reference.

"*A plot showing the scalability of the code for a medium-size system is desired, for instance, time (or speedup) vs. nr. of MPI ranks. "

We thank the reviewer for his/her advice.  We have added such analyses for QM/MM Molecular dynamics simulations and of QM/MM electron dynamics simulations (section 2.3.1 and 2.3.3 respectively). Two new figures (3 and 4) have been added to illustrate performance and scalability of the code.

"*Is the present version a pure MPI implementation or is that a hybrid MPI-OpenMP one? "

deMon2k is currently implemented following the MPI paradigm. We have specified this information in Introduction with reference to original publication (Ref. 24).

"*Something that is not clearly mentioned is what differentiates the present software from other existing packages for performing QM/MM simulations? This could be mentioned in the introduction and also in the conclusion sections. "

We have supplemented Introduction with a list of common computer programs enabling QM/MM simulations with appropriate references. In Introduction it is stated that:

" The latter (deMon2k) permits remarkably fast evaluation of energies, potentials and properties. (...) It (deMon2k) is therefore a very promising basis for conducting hybrid QM/MM simulations with DFT as the electronic-structure method."

"deMon2k also provides one of the very few implementations for conducting attosecond electron dynamics within polarizable MM environments"

"*Although QM/MM simulations for watery environments are suitable for testing the performance of the software, a catalytic reaction in an enzymic environment would be more interesting for realistic applications."

We fully agree with the reviewer, but for the sake of keeping our applicative tutorial easy to reproduce we have prefered to focus on QM/MM simulations in water. We don't see obstacle to investigate large biomolecules like enzymes. We have added a sentence in that sense in Conclusion: " To keep simple tutorial we have focused on small systems embedded in water. There is however no technical to tackle QM/MM simulation of more complex molecular systems like proteins or DNA.  "

"*In the manuscript, it is mentioned that "Tutorials with step-by-step descriptions of the simulations can be found on the deMon2k website". The reviewer couldn't find that set of tutorials. Thus, we recommend the authors to provide a direct link to the tutorials they referred to. "

The author is right. Actually, as stated in our cover letter the tutorials will be put on-line on the deMon2k webpage after publication of our article. For illustration we have deposited four of them a Supplementary material"

"*CHARMM22 topology/parameters set is obsolete, it is recommended to support instead CHARMM36 instead."

The reviewer is right. We have implemented CHARMM36 yet essentially because of lack of time. We have added a note in the text in Conclusion indicating that we are working on the inclusion of better force fields.

"*A more standard SN2 reaction of Cl+CH3Cl which has been reported previously in the literature would be instructive at this point. In this way, the reader could see how the methods implemented in the present work perform. "

We have chosen to show an application with non-symmetric SN2 reactions instead of the symmetric Cl+CH3Cl reaction proposed by the referee, because we it is easier to obtain “qualitatively” correct information for such reactions. For symmetric reactions, it would be necessary to make sure that the sampling is large enough to recover a symmetric free energy profile. Our purpose in this paper is not to study in great detail a given system but rather to propose applications that could be run with a limited amount a computational resource as a tutorial. We thus preferred to use reactions displaying a quite large reaction free energy.

"*A section with all novel features of the software in the form of a list of a table is desired. "

We thank the Reviewer for this good suggestion. We have added  such a Table in introduction (Table 1).

"In addition to the comments mentioned above, the reviewer detected several "not found" references throughout the text."

This problem apparently appeared at the creation of the files sent to reviewers by the server. Calls to Figures and Tables ("References") appear well in our word file. We mentioned this problem to the Editor rapidly after submission.

Round  2

Reviewer 2 Report

The authors addressed all my comments in a concise manner. Now, the readers can get an idea of both numerical accuracy/precision and performance scalability of this software.